# Simulation of NMR spectra at zero and ultralow fields from A to Z – a tribute to Prof. Konstantin L'vovich Ivanov

**Quentin Stern[1] and Kirill Sheberstov[2]**

[1]Univ Lyon CE1, CNRS, ENS Lyon, UCBL, Université de Lyon,
CRMN UMR 5280, 69100 Villeurbanne, France
[2]Laboratoire des biomolécules (LBM), Département de chimie, École normale supérieure,
PSL University, Sorbonne Université, CNRS, 75005 Paris, France CE2

**Correspondence:** Quentin Stern (quentin.stern@protonmail.com)

**Abstract.** TS1 Simulating NMR experiments may appear mysterious and even daunting for those who are new to the field. Yet, broken down into pieces, the process may turn out to be easier than expected. Quite the opposite, it is in fact a powerful and playful means to get insights into the spin dynamics of NMR experiments. In this tutorial paper, we show step by step how some NMR experiments can be simulated, assuming as little prior knowledge from the reader as possible. We focus on the case of NMR at zero and ultralow fields, an emerging modality of NMR in which the spin dynamics are dominated by spin–spin interactions rather than spin–field interactions, as is usually the case with conventional high-field NMR. We first show how to simulate spectra numerically. In a second step, we detail an approach to construct an eigenbasis for systems of spin-1/2 nuclei at zero field. We then use it to interpret the numerical simulations. In this attempt to make NMR simulation approachable, the authors wish to pay tribute to Prof. Konstantin L'vovich Ivanov, a great scientist and pedagogue who passed away on 5 March 2021. TS2

## 1 Introduction

NMR spectroscopists know well the advantages of performing experiments at the highest possible magnetic field. Increasing magnetic field strength boosts the sensitivity thanks to higher Boltzmann nuclear polarization and higher Larmor frequency (provided the signal linewidth is maintained constant). In addition to this already convincing advantage, higher magnetic fields also imply a larger frequency shift dispersion and therefore easier resolution of individual resonances in crowded spectra. This has motivated the use of ever-increasing magnetic fields (Thayer and Pines, 1987; Schwalbe, 2017; Wikus et al., 2022). The past year has witnessed the implementation of the first spectrometers operating at no less than 28 T, corresponding to a [1]H Larmor frequency of 1.2 GHz. (Wikus et al., 2022) There is no doubt that these new instruments will allow for unprecedented applications.

On the fringe of these great achievements, growing interest is pioneering the opposite strategy, namely, zero- to ultralow-field (ZULF) NMR, a modality of NMR experiments where the dominant interactions are spin–spin interactions rather than spin–field interactions (Thayer and Pines, 1987; Weitekamp et al., 1983; Blanchard and Budker, 2016; Blanchard et al., 2021; Tayler et al., 2017; Jiang et al., 2021). To realize such conditions, ZULF experiments are not performed in magnets but rather in mumetal magnetic shields that screen magnetic fields originating from the Earth and other surrounding sources, bringing the residual field down to nanotesla (nT) values. In this paper, "zero field" (ZF) designates the regime where heteronuclear spin–spin interactions dominate over spin–field interactions (Zeeman interactions), and the residual spin–field interactions are small enough for the Larmor period to be much longer than the coherence time (Blanchard and Budker, 2016). When this condition is met, decreasing the residual field to even lower values leaves the

NMR spectrum unchanged. "Ultralow field" (ULF) designates the regime where the spin–field interactions can be treated as a perturbation to the heteronuclear spin–spin interactions. This typically corresponds to fields on the order of tens to hundreds of nanotesla (Ledbetter et al., 2011). Liquid-state ZULF experiments result in $J$ spectra CE3 which do not feature any chemical shift information (Ledbetter et al., 2009). The regime where the intensity of heteronuclear spin–spin interactions is on the order of that of the spin–field interactions occurs typically in the range of microtesla (μT) to tens of microtesla and is referred to as Earth-field NMR (EF-NMR) (Callaghan and Le Gros, 1982; Appelt et al., 2006).

In the simplest form of ZULF experiments, the sample is thermally prepolarized in a permanent magnet (typically 2 T) (Tayler et al., 2017) and subsequently shuttled into the magnetic shields for detection at ZF or ULF. Alternatively, ZULF experiments may be coupled with hyperpolarization techniques (Theis et al., 2012; Butler et al., 2013b; Barskiy et al., 2019; Picazo-Frutos et al., 2023). In particular, parahydrogen-induced polarization (PHIP) has become common as a method for enhancing ZULF signal (Theis et al., 2011, 2012; Butler et al., 2013b). Once the sample is prepolarized (or hyperpolarized), coherences are excited using constant magnetic field pulses rather than radiofrequency (RF) pulses and are usually detected using optically pumped magnetometers (OPMs) rather than inductive coils (Ledbetter et al., 2009). Contrary to high-field instruments, ZULF spectrometers have the advantage of being cheap and relatively easy to assemble (Tayler et al., 2017). They are small enough to sit on a bench and do not require the use of cryogenics (at least if OPMs are used for detection).

Most people who have been introduced to the theory of high-field NMR have first encountered the vector model. The representation of a single-spin system as a vector in 3D space is a powerful tool to build intuition on what happens during an NMR experiment. Then, in a second step, the product operator formalism is necessary to understand the outcome of experiments involving interacting spins. At ZULF, couplings between spins need to be taken into account even to describe the simplest experiment, which consists of detecting the coherence between the singlet $S_0$ and triplet $T_0$ states of a pair of $J$-coupled heteronuclei, e.g., $^1$H and $^{13}$C (Blanchard and Budker, 2016). Polarization oscillates back and forth from one heteronucleus to the other, producing an observable oscillating signal whose frequency is given by the $J$ coupling between the two spins. The outcome of the experiment is simple – a single line at the $J$-coupling frequency – although it cannot be predicted by the vector model of high-field NMR and Bloch equations. Nonetheless, it is possible to build intuition regarding ZULF experiments in several ways. First, when dealing with two-spin systems, one can define spin operators at ZF in analogy to that at high field so as to translate some of the intuitions from high field to ZULF (Blanchard and Budker, 2016; Butler et al., 2013b). Second, there is a strong analogy between the energy states of electronic

spins in atoms and coupled nuclei at ZF (Butler et al., 2013a; Theis et al., 2013). The formalism of addition of angular momenta (widely used in atomic physics and rotational spectroscopy but less frequently in liquid-state NMR) can therefore be used to describe ZULF experiments. Finally, ZULF experiments can be numerically simulated easily, and – as is the case for high-field NMR – simulation provides a playful means to understand NMR experiments (Blanchard et al., 2020; Put et al., 2021). This tutorial paper is focused on the last two approaches.

We present a step-by-step procedure to numerically simulate ZULF spectra in some simple cases. The process is broken down into the following steps:

1. define the experimental sequence

2. define the spin system

3. compute the spin Hamiltonian

4. define the initial state – compute the initial density matrix

5. propagate the density matrix under the Hamiltonians

6. extract expectation values from the propagation

7. Fourier transform the expectation values to obtain a spectrum.

We assume that the reader is familiar with general concepts of NMR and that they are not necessarily used to performing spin dynamics simulations. We take particular care to detail the technical "tricks" which are generally omitted in research papers but are nonetheless essential to performing successful simulations. We present simulated spectra for XA$_n$ spin systems with $n$ between 1 and 5 with several excitation schemes. The spectra are simulated using MATLAB live scripts, which are available in the Supplement. The code is abundantly commented and is constructed so as to follow precisely the recipe presented in this paper. Each object and operation presented in this paper can thus be related to lines in the MATLAB code, and vice versa. PDF versions of the live scripts are available. We strongly advise the reader to read the code in parallel to the paper. In a second step, we interpret the simulated results by performing an analytical analysis of the XA$_n$ system using a theoretical framework coming from atomic physics. We show how to construct an eigenbasis and find the selection rules for the allowed transitions. This section is also supported with a code written in Wolfram Mathematica and with a step-by-step link between the text and lines in the code supporting the derived equations.

The reader might wonder whether it makes sense to go through all the details of simulating NMR experiments from scratch while there are powerful simulation packages which are freely available. SpinDynamica (Bengs and Levitt, 2018)

and Spinach (Hogben et al., 2011), which run on Mathematica and MATLAB, respectively, are probably the most appropriate tools for simulations at ZULF. The people who have programmed these have already gone through the hurdles of making them efficient and versatile for us, and they even provide code examples for the simulation of NMR spectra at ZULF.[1] However, it is the authors' opinion that performing simple simulations from scratch is the best way to get familiar with the quantum mechanical objects of NMR theory. Once one is confident with these objects and their language, one will make the best use of powerful packages such as SpinDynamica and Spinach. We note that several PhD theses from Pines' group at the Massachusetts Institute of Technology (MIT) present simulations of NMR spectra at ZULF (Theis, 2012; Blanchard, 2014; Sjolander, 2017). These theses contain code examples and are a useful resource for the beginner.

In writing this paper, the authors wish to pay tribute to their regretted lecturer and mentor Prof. Konstantin L'vovich Ivanov, known as Kostya by many, who was taken by COVID-19 on 5 March 2021 (Yurkovskaya and Bodenhausen, 2021). Kirill Sheberstov TS4 had Konstantin Ivanov as a PhD co-supervisor, performing research on long-lived states, parahydrogen-induced polarization, and chemically induced dynamic nuclear polarization (CIDNP). Konstantin Ivanov's deep understanding of the underlying physics allowed his group to work in very different directions, e.g., to combine CIDNP and ZULF NMR. During his PhD in Sami Jannin's team in Lyon, France, Quentin Stern collaborated with Konstantin Ivanov on a research project. In the course of their collaboration, Konstantin Ivanov gave Quentin Stern guidance on how to simulate experiments at ZF. A few pieces of advice turned into precious teachings for Quentin Stern. Sadly, these teachings were brutally interrupted by Konstantin Ivanov's death. Konstantin Ivanov's kindness and availability to give help and advice will forever remain an example for Quentin Stern and Kirill Sheberstov.

## 2 Theory – numerical simulation of spin dynamics

### 2.1 Define the experimental sequence

Most 1D NMR experiments can be broken down into three steps:

preparation − mixing − detection.

During the preparation, some nuclear polarization is acquired by letting the sample rest in a strong magnetic field (in most conventional experiments). Mixing consists of bringing the system to a non-stationary state whose oscillations are recorded during detection. In common high-field NMR

experiments, all the steps are performed in a strong magnet with a nearly constant magnetic field. Nuclear polarization is spontaneously acquired due to the high magnetic field, and both the mixing and detection are performed through the same RF coil using Faraday induction. At ZULF, there is no nuclear polarization, so the preparation has to be performed in different conditions. A common method is to shuttle the sample between a region of high field and a region of ZULF.

Figure 1 shows a typical experimental setup. A permanent magnet is used to prepolarize the sample, which is connected with the magnetic shields by a guiding solenoid coil. This coil ensures that the sample experiences a magnetic field with constant direction and sufficient strength during the transfer from the region of high field to inside the magnetic shields (i.e., the coil ensures an adiabatic transfer). Once the sample arrives in the magnetic shields at the location of detection, the Helmholtz coil continues to produce a magnetic field in the same direction as the solenoid, and the spin system is still distributed into Zeeman populations (Blanchard and Budker, 2016; Tayler et al., 2017). All the steps detailed until here are part of the preparation. In practice, the guiding solenoid and the Helmholtz coil produce a magnetic field which is much weaker than the prepolarizing magnet. However, this will not be taken into account in the simulation: we consider that the sample spends enough time in the prepolarizing magnet to reach Boltzmann equilibrium and that the transfer is sufficiently fast for us to neglect the change in polarization during the transfer.

A further step can optionally be added to the preparation, which consists of ramping down the magnetic field produced by the Helmholtz coil to bring the spins adiabatically to ZF. We will refer to experiments which include or do not include this step as the adiabatic field drop and sudden field drop experiments, respectively. In the case of sudden field drop experiments, the mixing step simply consists of switching off the magnetic field non-adiabatically (that is, fast enough to be considered instantaneous with respect to the evolution of the spin system). In the case of adiabatic field drop experiments, the sample is already at ZF at the end of preparation, so populations have to be mixed by applying a magnetic field pulse before any signal can be detected. This is analogous to high-field pulses except that it uses constant magnetic field rather than RF pulses. After the mixing, the oscillating magnetic field generated by the sample is detected by an optical magnetometer. In Fig. 1, the magnetometer is represented below the sample, that is, aligned along the $z$ axis with respect to the sample. We assume that the OPM is configured so as to be sensitive to magnetic field along the $z$ axis and that the spins are initially prepolarized along the same axis. Defining this axis as $z$ is a natural choice for high-field NMR spectroscopists, but note that other conventions exist (see, for example, Ledbetter et al., 2011). During detection, a weak magnetic field may be applied, either along the $z$ axis or along an orthogonal axis. In the latter case, the experiment is said to be performed under the ULF regime. In the absence of

---

[1]See, for example, http://spindynamics.org/wiki/index.php?title=Zerofield.m (last access: TS3).

applied magnetic field (and provided the residual magnetic field is properly zeroed at the location of the sample), the experiment is said to be performed under the ZF regime.

In summary, there are several possible combinations of experimental schemes. All of them start with prepolarizing the spins at high magnetic field. After the sample is transported into the magnetic shields, the field is dropped either suddenly or adiabatically, in which case a magnetic field pulse is applied. Finally, the oscillating magnetic field produced by the sample is detected along the $z$ axis, with or without a weak magnetic field applied along the $x$ axis. In the remaining of the paper, these sequences presented in Fig. 3b will be broken down into the following steps:

1. prepolarization,

2. transfer and coherence excitation, and

3. detection.

## 2.2 Define the spin system

This step consists of listing the different magnetic sites of the molecule whose ZULF spectrum is to be simulated and the interactions which the spins are subject to. This paper is concerned with small molecules in the liquid state. As is the case for high-field NMR, dipolar interactions are averaged out by rapid molecular tumbling and need not be taken into account (except as stochastic perturbation if one intends to include relaxation effects). Therefore, only the $J$ coupling and the Zeeman interactions are considered here.

In this paper, we consider spin systems of the form $XA_n$, where X is a $^{13}C$ spin coupled to $n$ equivalent $^1H$ spins A through a coupling $J_{AX}$. The A spins are coupled together through $J_{AA}$.

## 2.3 Compute the spin Hamiltonian

The Hamiltonian is the operator which represents the total energy of the system. Information about the spin system is mathematically encoded in the spin Hamiltonian. We will first present how the Hamiltonian for the Zeeman interaction of a single spin is computed based on Pauli matrices. Then, we will present the construction of a two-spin system using the Kronecker product of individual spin spaces to compute the Zeeman and the $J$-coupling Hamiltonians. Finally, we will show how the procedure is extended to an arbitrary number of spins.

Let us first assume that the system contains a single spin-1/2 interacting with the magnetic field $\boldsymbol{B}$ along the $z$ axis. The state of any spin system can be represented as a linear combination of basis vectors, which are called "kets" in Dirac's notation and are represented by the symbols $| \ \rangle$. For a single spin-1/2, two basis kets are necessary to represent the state of the system. We chose to represent the spin sys-

tem in the Zeeman basis:

$$B_Z^2 = \{ |\alpha\rangle , |\beta\rangle \} = \left\{ \begin{pmatrix} 1 \\ 0 \end{pmatrix}, \begin{pmatrix} 0 \\ 1 \end{pmatrix} \right\}. \tag{1}$$

The $|\alpha\rangle$ and $|\beta\rangle$ states correspond to the spin being parallel and antiparallel with the magnetic field, respectively. The general state in which the spin may be found is a linear combination of these two basis states. Because these states and their associated kets form a basis, their vector representation have the canonical form with only 0 and 1 coefficients. The space spanned by these two vectors is called a "Hilbert space" and has dimension 2, as indicated by the superscript in $B_Z^2$. Note that the choice of the Zeeman basis is convenient for numerical simulation but is not necessary. For example, one may use the coupled basis, which will be presented and used in Sect. 4. The same basis may be used to perform simulations at high field or ZULF, although one particular basis might turn out to be more convenient.

The angular momentum of a single spin is associated with the spin angular momentum operators, which can be represented as a vector with three Cartesian components:

$$\hat{I} = \left( \hat{I}_x , \hat{I}_y , \hat{I}_z \right). \tag{2}$$

These operators act on the Zeeman states in certain ways, e.g., $\hat{I}_x |\alpha\rangle = \frac{1}{2} |\beta\rangle$. To summarize the set of rules, it is convenient to use the matrix representation of the operators, with the matrix elements determined by the action of the operator on the $|\alpha\rangle$ and $|\beta\rangle$ states: $I_\mu^{rs} = \langle r | \hat{I}_k | s \rangle$, where $r, s \in \{\alpha, \beta\} ; \mu \in \{x, y, z\}$. This definition makes use of $\langle \ |$, i.e., the "bra", an object which is complementary to the ket and corresponds to the "Hermitean conjugate" of the ket. In the matrix representation of quantum mechanics, the Hermitean conjugate of a ket corresponds to the complex transpose of the vector representing the ket. The matrix representation of operators in quantum mechanics is very important for performing simulations, as they are constructed in such a way that any state or operation with the quantum system can be represented using linear algebra. The matrix representations of the three Cartesian components of the spin angular momentum operators are proportional to Pauli matrices $\hat{\sigma}_x$, $\hat{\sigma}_y$, and $\hat{\sigma}_z$:

$$\hat{I}_x = \frac{1}{2}\hat{\sigma}_x = \frac{1}{2} \begin{pmatrix} 0 & 1 \\ 1 & 0 \end{pmatrix},$$
$$\hat{I}_y = \frac{1}{2}\hat{\sigma}_y = \frac{1}{2} \begin{pmatrix} 0 & -i \\ i & 0 \end{pmatrix},$$
$$\hat{I}_z = \frac{1}{2}\hat{\sigma}_z = \frac{1}{2} \begin{pmatrix} 1 & 0 \\ 0 & -1 \end{pmatrix}. \tag{3}$$

The interaction of a single spin with a magnetic field $\boldsymbol{B}$ is given by the Zeeman Hamiltonian:

$$\hat{H}_Z = -\gamma \, \boldsymbol{B} \cdot \hat{I} = -\gamma \left( B_x \hat{I}_x + B_y \hat{I}_y + B_z \hat{I}_z \right), \tag{4}$$

**(a)** Scheme of the experimental setup          **(b)** Scheme of the experimental sequences

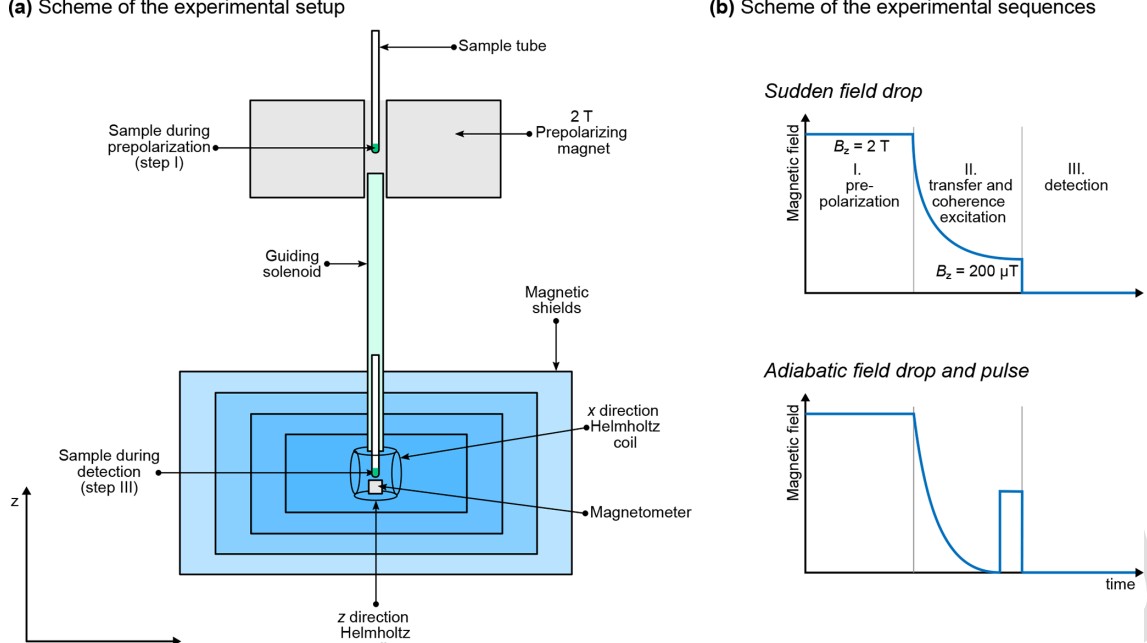

**Figure 1. (a)** Typical experimental setup for ZULF experiments. Note that the sample is represented in two places in the same drawing even if there is a single sample. **(b)** Schemes of the experimental sequences for measurements at ZF using a sudden field drop or an adiabatic field drop followed by a pulse of static magnetic field. TS5

where $\gamma$ is the gyromagnetic ratio of the spin (in rad s$^{-1}$ T$^{-1}$). The dot product of the vectors of the magnetic field and of the spin angular momentum (vectors and vector operators are denoted in bold throughout the text) is expanded on the right member of Eq. (4). Note that we have omitted the reduced Planck constant $\hbar$ in Eq. (4), which implies that the energy is expressed in radians per second (rad s$^{-1}$) rather than in joules. This is the case throughout this paper. In many cases, the magnetic field is aligned with one of the axes. If it points along the $z$ axis, i.e., $\boldsymbol{B} = \begin{pmatrix} 0 & 0 & B_0 \end{pmatrix}$, Eq. (4) simplifies to

$$\hat{H}_Z = -\gamma B_0 \hat{I}_z = \omega^0 \hat{I}_z = \frac{1}{2} \begin{pmatrix} +\omega^0 & 0 \\ 0 & -\omega^0 \end{pmatrix}, \quad (5)$$

where $\omega^0 = -\gamma B_0$ is the Larmor frequency of the spin. This expression is valid regardless of the intensity of the magnetic field, i.e., at high field as well as at ZULF. The Zeeman states, $|\alpha\rangle$ and $|\beta\rangle$, which correspond to the spin being parallel and antiparallel with the magnetic field, respectively, are eigenstates of the Zeeman Hamiltonian; that is, they satisfy the relations $\hat{H}_Z |\alpha\rangle = +1/2 |\alpha\rangle$ and $\hat{H}_Z |\beta\rangle = -1/2 |\beta\rangle$. Eigenstates of a Hamiltonian are of particular importance; they are states which do not evolve under the effect of that Hamiltonian (ignoring the accumulation of the phase factor, which turns out to be irrelevant in most of the experiments), i.e., stationary states.

The single spin whose Hamiltonian is given by Eq. (5) lives in a Hilbert space of dimension 2. To represent a pair of spins of 1/2, we need to use a Hilbert space with a dimension of 4. To do so, we redefine the angular momentum operators in this higher-dimension space. The matrix representations of the angular momentum operators $\hat{I}_{1x}$, $\hat{I}_{1y}$, and $\hat{I}_{1z}$ and $\hat{I}_{2x}$, $\hat{I}_{2y}$, and $\hat{I}_{2z}$ of spin 1 and spin 2, respectively, are given by the Kronecker product of matrices of single-spin angular momentum operator and the identity operator, in the appropriate order. For the $z$-axis angular momentum operators, we have

$$\hat{I}_{1z} = \hat{I}_z \otimes \hat{1} = \frac{1}{2} \begin{pmatrix} 1 & 0 \\ 0 & -1 \end{pmatrix} \otimes \begin{pmatrix} 1 & 0 \\ 0 & 1 \end{pmatrix}$$

$$= \frac{1}{2} \begin{pmatrix} +1 & 0 & 0 & 0 \\ 0 & +1 & 0 & 0 \\ 0 & 0 & -1 & 0 \\ 0 & 0 & 0 & -1 \end{pmatrix} \quad (6)$$

and

$$\hat{I}_{2z} = \hat{1} \otimes \hat{I}_z = \begin{pmatrix} 1 & 0 \\ 0 & 1 \end{pmatrix} \otimes \frac{1}{2} \begin{pmatrix} 1 & 0 \\ 0 & -1 \end{pmatrix}$$

$$= \frac{1}{2} \begin{pmatrix} +1 & 0 & 0 & 0 \\ 0 & -1 & 0 & 0 \\ 0 & 0 & +1 & 0 \\ 0 & 0 & 0 & -1 \end{pmatrix}. \quad (7)$$

Similar expressions are obtained for the matrices of $x$ and $y$ operators. They are not shown here but are available in many textbooks (Hore et al., 2015; Levitt, 2013). Here, we have

used the following convention for the Kronecker product

$$
\begin{pmatrix} a & b \\ c & d \end{pmatrix} \otimes \begin{pmatrix} \alpha & \beta \\ \gamma & \delta \end{pmatrix}
$$

$$
= \begin{pmatrix} a \begin{pmatrix} \alpha & \beta \\ \gamma & \delta \end{pmatrix} & b \begin{pmatrix} \alpha & \beta \\ \gamma & \delta \end{pmatrix} \\ c \begin{pmatrix} \alpha & \beta \\ \gamma & \delta \end{pmatrix} & d \begin{pmatrix} \alpha & \beta \\ \gamma & \delta \end{pmatrix} \end{pmatrix}
$$

$$
= \begin{pmatrix} a\alpha & a\beta & b\alpha & b\beta \\ a\gamma & a\delta & b\gamma & b\delta \\ c\alpha & c\beta & d\alpha & d\beta \\ c\gamma & c\delta & d\gamma & d\delta \end{pmatrix}. \tag{8}
$$

The two operators defined by Eqs. (6) and (7) are the same as the one given by Eq. (5), except that the world of spin 1 now contains spin 2, and vice versa. This representation corresponds to a basis that is the Kronecker product of the basis of the individual spins.

$$
B_Z^4 = B_Z^2 \otimes B_Z^2 = \{ |\alpha\alpha\rangle , |\alpha\beta\rangle , |\beta\alpha\rangle , |\beta \beta\rangle \} \tag{9}
$$

For the case where the magnetic field points along the $z$ axis, the total Zeeman Hamiltonian for the two spins can now be computed using Eq. (5) in the basis of Eq. (9) as the sum of the two Zeeman Hamiltonians:

$$
\hat{H}_Z = \hat{H}_{Z,1} + \hat{H}_{Z,2} = \omega_1^0 \hat{I}_{1z} + \omega_2^0 \hat{I}_{2z}
$$

$$
= \frac{1}{2} \begin{pmatrix} +\omega_1^0 + \omega_2^0 & 0 & 0 & 0 \\ 0 & +\omega_1^0 - \omega_2^0 & 0 & 0 \\ 0 & 0 & -\omega_1^0 + \omega_2^0 & 0 \\ 0 & 0 & 0 & -\omega_1^0 - \omega_2^0 \end{pmatrix}, \tag{10}
$$

where $\omega_1^0$ and $\omega_2^0$ are the Larmor frequencies of spin 1 and 2, respectively. Note that in a Hilbert space of several spins, it is useful to define projections of total angular momentum operators:

$$
\hat{I}_x = \hat{I}_{1x} + \hat{I}_{2x},
$$
$$
\hat{I}_y = \hat{I}_{1y} + \hat{I}_{2y},
$$
$$
\hat{I}_z = \hat{I}_{1z} + \hat{I}_{2z}. \tag{11}
$$

Note that these operators are represented by the same symbol as their equivalent in the single-spin Hilbert space (see Eq. 3). It should be clear from the context whether the operator corresponds to a single-spin or multiple-spin Hilbert space. Where confusion may remain, we will indicate the dimension of the space on which the operator acts.

At this point, the two spins are represented in a common space, but they do not interact. The $J$-coupling Hamiltonian for the pair of spins is given by

$$
\hat{H}_J = 2\pi J \hat{I}_1 \cdot \hat{I}_2 = 2\pi J \left( \hat{I}_{1x} \hat{I}_{2x} + \hat{I}_{1y} \hat{I}_{2y} + \hat{I}_{1z} \hat{I}_{2z} \right)
$$

$$
= \pi J \begin{pmatrix} 1/2 & 0 & 0 & 0 \\ 0 & -1/2 & 1 & 0 \\ 0 & 1 & -1/2 & 0 \\ 0 & 0 & 0 & 1/2 \end{pmatrix}, \tag{12}
$$

where $J$ is the $J$ coupling between the two spins (in Hz). Compared with the Zeeman Hamiltonian (see Eq. 10), the $J$-coupling Hamiltonian has the particularity to have off-diagonal elements in the $\{ |\alpha\beta\rangle , |\beta\alpha\rangle \}$ subspace, which implies that the $J$ interaction mixes the $|\alpha\beta\rangle$ and $|\beta\alpha\rangle$ states. In other words, due to the $J$ interaction, these two states are no longer eigenstates of the spin system.

In the case of a system of $n$ spins of $1/2$, the same procedure can be applied to define the angular momentum operators and the Hamiltonians. These operators can be represented as $2^n \times 2^n$ matrices. Their Zeeman basis can be constructed as in Eq. (9), taking all possible combinations of $|\alpha\rangle$ and $|\beta\rangle$ states of the individuals spins. Equations (6) and (7) generalize to

$$
\hat{I}_{kz} = \otimes_{l=1}^n \hat{u}_{lz}^{2\times2} \text{ where } \hat{u}_{lz}^{2\times2} = \begin{cases} \hat{1}^{2\times2} \text{ if } l \neq k, \\ \hat{I}_z^{2\times2} \text{ if } l = k, \end{cases} \tag{13}
$$

where $\hat{1}$ and $\hat{I}_{kz}$ are the identity operator and the $z$ angular momentum operator of spin $k$ in an $n$-spin Hilbert space, and the $\hat{u}_{lz}^{2\times2}$ operator is defined in a single-spin Hilbert space. The $z$ projection of total angular momentum operators is given by

$$
\hat{I}_z = \sum_{l=1}^n \hat{I}_{lz}. \tag{14}
$$

Equations (13) and (14) are shown for $z$ operators but apply similarly for $x$ and $y$ operators. The Zeeman Hamiltonian for a system of $n$ spins is given by

$$
\hat{H}_Z = -\sum_{l=1}^n \gamma_l \boldsymbol{B} \cdot \hat{I}_l
$$

$$
= -\sum_{l=1}^n \gamma_l \left( B_x \hat{I}_{lx} + B_y \hat{I}_{ly} + B_z \hat{I}_{lz} \right), \tag{15}
$$

where $\gamma_l$ is the gyromagnetic ratio of spin l. The $J$ Hamiltonian in the same space is given by

$$
\hat{H}_J = 2\pi \sum_{l>k}^n J_{lk} \hat{I}_l \cdot \hat{I}_k
$$

$$
= 2\pi \sum_{l>k}^n J_{lk} \left( \hat{I}_{lx} \hat{I}_{kx} + \hat{I}_{ly} \hat{I}_{ky} + \hat{I}_{lz} \hat{I}_{kz} \right), \tag{16}
$$

where $J_{lk}$ is the $J$ coupling between spins $l$ and $k$ (in Hz). Because a spin is not $J$ coupled to itself, the sum in Eq. (16) does not include terms with $l = k$. Furthermore, to avoid counting terms twice, terms with $l < k$ are not included either, leaving only $l > k$ terms. The expression of the Zeeman Hamiltonian and $J$ Hamiltonian in Eqs. (15) and (16), respectively, are valid both at high field and at ZULF. What makes the difference between the two regimes is the relative intensity of the two contributions.

## 2.4 Define the initial state – compute the initial density matrix

The state of a spin system during an NMR experiment is described by a density operator. If $|\psi\rangle$ is a ket representing the state of the system as a linear combination of basis states (like those defined in Eqs. 1 and 9), the density operator is given by

$$\hat{\rho} = \overline{|\psi\rangle \langle\psi|}, \tag{17}$$

where the upper bar represents the ensemble average over all identical spin systems in the sample – the operation performed by the density operator. This averaging makes the density operator formalism well-suited for NMR, where the experiment consists of observing a large number of identical spin systems at the same time rather than a single spin system. The matrix representation of the density operator (and of any other spin operator) is achieved by calculating all the matrix elements $\rho_{rs} = \langle r|\hat{\rho}|s\rangle$, where $|r\rangle$ and $|s\rangle$ are basis states. For example, the matrix representation of the density operator for the $|\alpha\rangle$ and $|\beta\rangle$ states of a single spin yields

$$\hat{\rho}_\alpha = \overline{|\alpha\rangle \langle\alpha|} = \begin{pmatrix} 1 \\ 0 \end{pmatrix} \begin{pmatrix} 1 & 0 \end{pmatrix} = \begin{pmatrix} 1 & 0 \\ 0 & 0 \end{pmatrix},$$

$$\hat{\rho}_\beta = \overline{|\beta\rangle \langle\beta|} = \begin{pmatrix} 0 \\ 1 \end{pmatrix} \begin{pmatrix} 0 & 1 \end{pmatrix} = \begin{pmatrix} 0 & 0 \\ 0 & 1 \end{pmatrix}. \tag{18}$$

To start a simulation, we need to determine the density matrix of the system at the initial point of the experiment. We assume that the sample has spent enough time in the prepolarizing magnet to reach thermal equilibrium; that is, the spin system follows Boltzmann's distribution of states. In this case, the density matrix is given by

$$\hat{\rho}_{eq} = \frac{\exp\left(-\frac{\hat{H}}{k_B T},\right)}{Z} \tag{19}$$

where $\hat{H}$, $k_B$, and $T$ are the Hamiltonian operator of the spin system, Boltzmann's constant, and the temperature, respectively. Operation exp( ) denotes the matrix exponentiation. Note that this operation does *not* consist of applying $f(x) = \exp(x)$ to each element of the matrix. It is a more complex operation, which is realized in MATLAB by the built-in function *expm* (rather than *exp*). $Z$ is a normalization constant, which ensures that the density matrix has unit trace. It is given by

$$Z = \mathrm{Tr}\left\{\exp\left(-\frac{\hat{H}}{k_B T}\right)\right\}. \tag{20}$$

The prepolarizing step of the experiments that we intend to simulate occurs in a strong magnetic field (in the sense that the Zeeman interaction is largely dominating all other interactions), as in a standard high-field experiment. In this case, we can compute the thermal equilibrium by taking only the Zeeman terms into account. For a single spin with Larmor frequency $\omega^0$ and gyromagnetic ratio $\gamma$ in prepolarizing field $B_p$, the thermal equilibrium density matrix yields

$$\hat{\rho}_{eq} = \frac{\exp\left(-\frac{\hbar\omega^0 \hat{I}_z}{k_B T}\right)}{Z} = \frac{\exp\left(+\frac{\hbar\gamma B_p \hat{I}_z}{k_B T}\right)}{Z}$$

$$= \begin{pmatrix} \frac{1+P}{2} & 0 \\ 0 & \frac{1-P}{2} \end{pmatrix} = \frac{1}{2}\hat{1} + P\hat{I}_z, \tag{21}$$

where $P$ is the polarization of the nucleus along the $z$ axis (for positive $\gamma$, it corresponds to the population excess of the $|\alpha\rangle$ state with respect to the $|\beta\rangle$ state), defined by

$$P = \tanh\left(\frac{\hbar\gamma B_p}{2k_B T}\right). \tag{22}$$

Note that the use of $\hbar$ in the expression of the Hamiltonian (i.e., expressing the energy in joules) cannot be avoided here, to ensure consistency of units. To obtain the expression on the right-hand side of Eq. (21), we have jumped several steps of calculation which are all based on the definition of polarization. This expression for the density matrix is exact for a spin whose only interaction is the Zeeman interaction, which we have assumed here.

For an $n$-spin system, we take the Kronecker product of density matrices of individual spins $\hat{\rho}_{eq,l}^{2\times2}$.

$$\hat{\rho}_{eq} \approx \otimes_{l=1}^n \hat{\rho}_{eq,l}^{2\times2} = \otimes_{l=1}^n \left(\frac{\hat{1}^{2\times2}}{2} + P_l \hat{I}_z^{2\times2}\right)$$

$$= \frac{\hat{1}}{2^n} + \frac{1}{2^{n-1}} \sum_{l=1}^n P_l \hat{I}_{lz} \tag{23}$$

The expression is approximate in the sense that it neglects all spin–spin interactions. This approximation is valid unless the system is highly polarized, which is the case even at very high field (without hyperpolarization). To avoid confusion, we specified that the operators $\hat{\rho}_{eq,l}^{2\times2}$, $\hat{1}^{2\times2}$, and $\hat{I}_z^{2\times2}$ act on a single-spin Hilbert space ($2 \times 2$ matrix). On the contrary, the operators $\hat{1}$ and $\hat{I}_{lz}$ act on spin states of $n$ spins, and accordingly their matrix representations have dimensionality of $2^n \times 2^n$ (for spins of $1/2$). As shown by Eq. (23), one may compute the density matrix either using the Kronecker product of operators in a single-spin Hilbert space or by summing the operators in a Hilbert space of $n$ spins.

In many textbooks (Hore et al., 2015; Levitt, 2013), one encounters simplified expressions of the density operator. First, it is common to remove the identity component:

$$\hat{\rho}_{eq} \rightarrow \hat{\rho}_{eq} - \frac{\hat{1}}{2^n}, \tag{24}$$

where $n$ is the number of spins in the system. Because all operators commute with the identity, this does not affect the result of propagation. The resulting expression is simpler

($\hat{\rho}_{eq} = P\hat{I}_z^{2\times2}$ for a single spin), which is convenient for calculations by hand. It may also make the numerical propagation faster and more precise. Another common simplification is to drop the polarization factor. For a single spin, the two combined simplifications yield

$$\hat{\rho}_{eq} \rightarrow \hat{I}_z. \tag{25}$$

Simplifications are useful, but they should be handled with care. The polarization factor $P$ is different for spins with different gyromagnetic ratio. If it is dropped without introducing further corrections, the relative sizes of the population of spins with different gyromagnetic ratios will not be respected. In the simulations presented here, we will compute the initial density matrix using the transformation of Eq. (24) but not that of Eq. (25).

## 2.5   Propagate the density matrix under the Hamiltonians

We have seen how to compute the initial density matrix and the matrix representation of the Hamiltonian. We now describe how the evolution of the system (represented by the density matrix) evolves with time under a given Hamiltonian. This will be used at several steps of the simulation: when the sample is brought adiabatically to ZF, during the pulse, and during the signal measurement.

The evolution of a quantum system with time is given by the time-dependent Schrödinger equation. Its equivalent for the evolution of density matrix is the Liouville–von Neumann equation $\frac{d}{dt}\hat{\rho}(t) = -i\left[\hat{H}(t), \hat{\rho}(t)\right]$, which has the solution

$$\hat{\rho}(t) = \hat{U}(t)\hat{\rho}_0\hat{U}^{-1}(t), \tag{26}$$

where $\hat{\rho}_0$ is the density matrix at $t = 0$, and $\hat{U}$ is the propagator during time $t$, which is defined as

$$\hat{U}(t) = \exp\left(-i\hat{H}t\right), \tag{27}$$

where $\hat{H}$ is the total Hamiltonian. The operation of Eq. (26) "takes" the spin system from $\hat{\rho}_0$ to $\hat{\rho}(t)$. Again, note that exp( ) denotes the matrix exponentiation and *not* element-by-element exponentiation. An important case of propagator is the rotation operator. For an angular momentum operator $\hat{I}_\mu$, with $\mu \in \{x, y, z\}$, the propagator $\exp\left(-i\hat{I}_\mu\theta\right)$ is called a rotation operator; it represents a rotation of the spins of angle $\theta$ around axis $\mu$, when applied to the density matrix using Eq. (26). For a single spin subject to a static magnetic field along the $z$ axis, the total Hamiltonian is the Zeeman Hamiltonian (see Eq. 5) which causes the spin to rotate around the $z$ axis; this rotation can be expressed using the rotation operator $\exp\left(-i\hat{H}t\right) = \exp\left(-i\omega_0\hat{I}_zt\right)$ with angle $\omega_0 t$.

The brute force calculation of the exponential operator in an arbitrary basis is computationally challenging as it requires calculating the Taylor expansion of the $\hat{H}$ operator. To avoid this, the calculation of the propagator (Eq. 27) is usually performed by diagonalizing the Hamiltonian and then taking the complex exponent for each of its eigenvalues, $\exp(-i\omega_k t)$, where $\omega_k$ denotes the $k$th eigenvalue. Therefore, the transformation to the eigenbasis of the Hamiltonian implicitly happens during most spin dynamics simulations, meaning that, even if it was not set by the user, this is likely done by the linear algebra packages of the software. One may note that the basis does not affect the result of the calculation, but the choice of a more appropriate one may help rationalize the problem. In many cases, the initial choice is the Zeeman basis, in which spin operators are readily introduced based on Kronecker products of the Pauli matrices. Depending on the symmetry of the problem, it might be more convenient to change the basis to another one. As we will see in Sect. 4.1, a choice of coupled basis is preferable for understanding zero-field $J$ spectroscopy of coupled spins.

It is important to remark that Eq. (27) is only valid if the Hamiltonian is constant during the evolution period. The case where the Hamiltonian is time dependent is treated below. Note that the propagator is a unitary operator and therefore has the convenient property that its inverse is equal to its complex transpose (i.e., $\hat{U}^{-1} = \hat{U}^\dagger$), which is much faster to compute than the matrix inverse $\hat{U}^{-1}$.

Equations (26) and (27) allow us to know the state of the system at any time $t$ from the initial time $t = 0$. To simulate the signal produced by the spin system during the course of the experiment, we must calculate the time domain signal at different time points. Note that in this case the Hamiltonian remains constant during free evolution. To calculate the signal at fixed time steps, it is convenient to first calculate the propagator $\hat{U}(dt)$ over period $dt$. We then apply Eq. (26) recursively to get the new density matrix $\hat{\rho}(t_{k+1})$ from the previous one $\hat{\rho}(t_k)$,

$$\hat{\rho}(t_{k+1}) = \hat{U}(dt)\hat{\rho}(t_k)\hat{U}^{-1}(dt), \tag{28}$$

where $t_{k+1} - t_k = dt$. To simulate ZULF spectra, we will also encounter situations where the Hamiltonian is time dependent. First, the Hamiltonian can vary with time but be "constant by block". This is, for example, the case for the sudden field drop; the system is under a certain Zeeman Hamiltonian in the beginning of the experiment and suddenly under the ZULF Hamiltonian during detection. This situation does not present particular difficulties; the evolution of the system can be described step by step by both Eqs. (26) and (28).

Second, the Hamiltonian can vary continuously, as in the case of the adiabatic field drop, where the intensity of the magnetic field is ramped down to zero. This event can be simulated by propagating the evolution of the system during time intervals which are sufficiently short for the Hamiltonian to be considered constant during this time interval. The propagator must then be computed for each time increment. The form of the equation for propagation is similar to Eq. (28),

$$\hat{\rho}(t_{k+1}) = \hat{U}(t_k \rightarrow t_{k+1})\hat{\rho}(t_k)\hat{U}^{-1}(t_k \rightarrow t_{k+1}), \tag{29}$$

where the propagator is given by

$$\hat{U}(t_k \to t_{k+1}) = \exp\left(-i\hat{H}(t_k)\mathrm{d}t\right), \tag{30}$$

where $\hat{H}(t_k)$ is the Hamiltonian at time $t_k$. Note that the choice of $\hat{H}(t_k)$ rather than $\hat{H}(t_{k+1})$ in Eq. (30) is arbitrary, but in the limit of small intervals, the choice has no consequence.

## 2.6 Extract expectation values from the propagation

The propagation procedure described above gives access to the density matrix along time. To simulate the time domain signal, we need to extract a physical quantity from the density matrix as it evolves with time. The measured physical quantity of a ZULF experiment is the magnetic field produced by the nuclear spins of the sample at the location of an OPM. In a first approximation, we can consider that the whole sample is a point dipole interacting with the OPM and that this total dipole is the sum of the dipoles of the individual spin systems (Fig. 2 gives a visual representation of the approximation). Whether this approximation is appropriate or not depends on the geometry of the experimental setup. We have chosen the $z$ axis as the quantization axis (defined by the detector, i.e., the OPM). Therefore, the physical quantity that we need to compute is the total magnetic field produced by the spins along the $z$ axis at the location of the vapor cell:

$$\langle B_z \rangle = \frac{\mu_0}{2\pi} \frac{\langle \hat{\mu}_z^{\mathrm{tot}} \rangle}{r^3} = \frac{\mu_0}{2\pi} \frac{N \langle \hat{\mu}_z \rangle}{r^3}, \tag{31}$$

where $\langle \hat{\mu}_z^{\mathrm{tot}} \rangle$, $\mu_0$, $N$, $\langle \hat{\mu}_z \rangle$, and $r$ are the magnetic moment of the sample along the $z$ axis, the permeability of free space, the number of identical spin systems in the sample, their individual magnetic moments along the $z$ axis, and the distance between the center of the sample and the center of the vapor cell, respectively.

For each identical spin system, we then compute the magnetic moment as the sum of the contributions of each spin $l$.

$$\langle B_z \rangle = \frac{\mu_0}{2\pi} \frac{N}{r^3} \sum_{l=1}^{n} \langle \hat{\mu}_{lz} \rangle = \frac{\mu_0}{2\pi} \frac{N\hbar}{r^3} \sum_{l=1}^{n} \gamma_l \langle \hat{I}_{lz} \rangle, \tag{32}$$

where $\hat{\mu}_{lz}$, $\gamma_l$, and $\hat{I}_{lz}$ are the magnetic moment, the gyromagnetic ratio, and the angular momentum along the $z$ axis of spin $l$, respectively. Note that $n$ and $N$ represent the number of spins in the molecule and the number of molecules in the sample, respectively. The notation $\langle \ \rangle$ denotes the expectation value of a quantity. Particularly important ones are those that can be physically measured in the experiment. In the density matrix formalism that we are using, the expectation value of a physical quantity related to an operator $\hat{A}$ is given by

$$\langle A \rangle = \mathrm{Tr}\left\{\hat{A}\hat{\rho}\right\}, \tag{33}$$

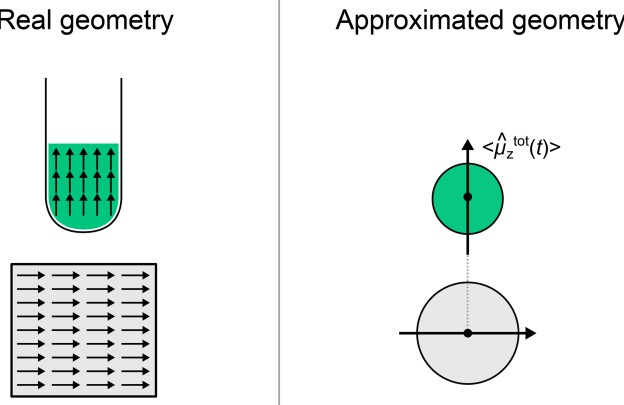

| Real geometry | Approximated geometry |

**Figure 2.** Comparison of the real geometry of the sample of the OPM with the approximated one. The arrows represent local magnetization vectors parallel to the total magnetization vector.

where $\mathrm{Tr}\{\ \}$ denotes the matrix trace, i.e., the sum of all diagonal elements of the matrix representation of the operator. Note that the expectation value of $\hat{\mu}_{lz}$ (or $\hat{I}_{lz}$) is proportional to the polarization level of spin $l$ which was accounted for in Eqs. (21) and (22). Therefore, the total magnetic moment calculated with Eq. (32) depends on the polarization of the different spin species.

If $\hat{\rho}(t)$ is the density matrix at time $t$, we obtain the signal $S(t)$ measured by the OPM by plugging Eq. (32) into Eq. (33)

$$S(t) = \langle B_z \rangle(t) = \frac{\mu_0}{2\pi} \frac{N\hbar}{r^3} \mathrm{Tr}\left\{\hat{O}\hat{\rho}(t)\right\}, \tag{34}$$

where we have defined a "detection operator",

$$\hat{O} = \sum_{l=1}^{n} \gamma_l \hat{I}_{lz}. \tag{35}$$

To obtain Eq. (34), we have used the fact that taking the trace of a matrix is a linear operation, and so the trace of a sum is the sum of the traces.

In the case of a sample with volume $V = 100\,\mu\mathrm{L}$ of $^{13}\mathrm{C}$-formic acid prepolarized at 2 T at 298 K, with molar mass of $46\,\mathrm{g\,mol^{-1}}$ TS8 and density of $1.22\,\mathrm{g\,mL^{-1}}$, one finds that the amplitude of the oscillating magnetic field generated by the sample at a distance of $r = 1\,\mathrm{cm}$ is on the order of $10\,\mathrm{pT}$ using the above equations. This estimation does not take into account demagnetization effects caused by distribution of spins in space, giving the upper limit for the expected field. Experimentally measured magnetic fields are about 10 times smaller (Tayler et al., 2017).

## 2.7 Fourier transform the expectation values to obtain a spectrum

The time domain signal is what is measured by the ZULF NMR spectrometer. The final step of the simulation is to

transform the measured signal from the time domain to the frequency domain using a discrete Fourier transform. Programming environments such as MATLAB or Mathematica are equipped with built-in functions for fast Fourier transformation. We will not discuss the mathematics behind this process, but we will give a few practical hints. Contrary to high-field NMR, ZULF spectra can be obtained with real magnetic field units (rather than arbitrary units). We will show how such units can be obtained.

Let us call $t$ and $S$ the arrays of numbers containing the time and corresponding time domain signal values, respectively, which resulted from the previous steps (note that, in MATLAB's programming environment, such arrays are usually called vectors). Let us call $K$ the number of elements of both arrays (which corresponds to the number of points in the time domain signal). For now, $S$ consists of a sum of oscillating signals which do not decay with time as our simulation did not include relaxation effects. If we perform a Fourier transform on $S$, we will obtain non-Lorentzian line shapes (with distinctive sinc patterns). We must therefore artificially include relaxation by multiplying the signal with an apodization function, to force the signal to decay to 0. For liquid-state signals, the most common choice is a monoexponential decay which can be expressed as

$$S'_k = S_k \exp(-\pi l_B t_k) = S_k \exp\left(-\frac{t_k}{T_2}\right), \qquad (36)$$

where $S_k$, $t_k$, $l_B$, and $T_2$ are the $k$th elements of $t$ and $S$, the line broadening (in Hz), and the coherence time constant (in s), respectively. Note that the coherence time constant is often referred to as the spin–spin relaxation constants or transverse relaxation time constant. The signal intensities $S'_k$ define the apodized signal array $S'$. As shown in Eq. (36), we may choose to express the apodization function using either the coherence time constant $T_2$ or the line broadening $l_B$ (in Hz), which are related by $\pi l_B = 1/T_2$. The former is the time constant at which the time domain signal decays, while the latter is the full width at half height (FWHH) of the signals. In order to avoid "truncating" the decay of the time domain signal and the related spectral artifacts, we must fulfill the condition $T_2 \ll t_{aq}$, where $t_{aq} = \max\{t_k\}$ is the acquisition time (or the length of the signal in the time domain). Typically, we may choose $T_2$ and $t_{aq}$ so that $t_{aq} = 5T_2$. Table 1 summarizes the parameters which were used in this paper.

The apodization function of Eq. (36) yields Lorentzian signals as one would expect. However, without further apodization, the baseline of the spectra will have some distortions (Zhu et al., 1993), with the main distortion being a small offset of the baseline. This problem arises because the time domain signal has its first point at time $t = 0$, so the Fourier transform gives the integral of the first segment of twice larger amplitude than it should be. As proposed by Otting, this baseline offset can be removed by weighting the first point of the time domain signal by factor $1/2$ (Otting et al., 1986). However, because the integral of the Fourier transform is proportional to the first point of the time domain signal, this apodization does not preserve the integral. To obtain spectra without baseline offset and preserving the integral, we propose to use an apodization function which weights all points by 2 expect for the first one:

$$S''_k = \begin{cases} S'_k & \text{if } k = 1, \\ 2S'_k & \text{otherwise.} \end{cases} \qquad (37)$$

We show in the Supplement that this apodization function preserves the integral (see Sect. S2.1 TS9).

In MATLAB programming language, the function for fast Fourier transformation *fft()* takes array $S'$ as input and returns the frequency domain array which corresponds to the simulated spectrum. Optionally, one may add a second argument $L$ to *fft()* to include a zero-filling CE4 in the Fourier transform. Including zero-filling has the advantage of increasing the number of points per FWHH on the spectrum without increasing the computation time of the propagation. Due to MATLAB's Fourier transform convention, it is convenient to retransform the signal with *fftshift()* in order to obtain a Fourier transformed signal with 0 as the middle frequency. We then divide the output of MATLAB's Fourier transform by the number of points $L$:

$$S(\nu) = \frac{1}{L} F S(t), \qquad (38)$$

where $F$ designates the Fourier transform. The frequency domain signal obtained after this whole procedure has units of magnetic field (e.g., pT). Changing the zero-filling $L$ changes the intensity of the frequency domain signal but preserves the integrals.

MATLAB's *fft()* function does not generate the frequency array associated with the Fourier transformed signal. The frequency array $\nu$ (in Hz) can be generated based on the following expression:

$$\nu_k = \frac{k}{L} f, \; k \in \left[\!\left[ -\frac{L}{2}; \frac{L}{2} - 1 \right]\!\right], \qquad (39)$$

where the sampling frequency (in Hz) is given by

$$f = \frac{K - 1}{t_{aq}}. \qquad (40)$$

The sampling frequency of the time domain signal gives the maximum frequency that can be appropriately sampled. Figure 3 illustrates the consequence of choosing a sampling frequency which is lower than the maximum frequency. If the sampling frequency is lower than the signal to be sampled, the Fourier transformed signal lies outside the spectral width (between $-f/2$ and $+f/2$). However, due to the "refolding" effect" of the Fourier transform, the signal still appears in the spectrum but at irrelevant positions. To avoid this, one may repeat the simulation by increasing the sampling frequency

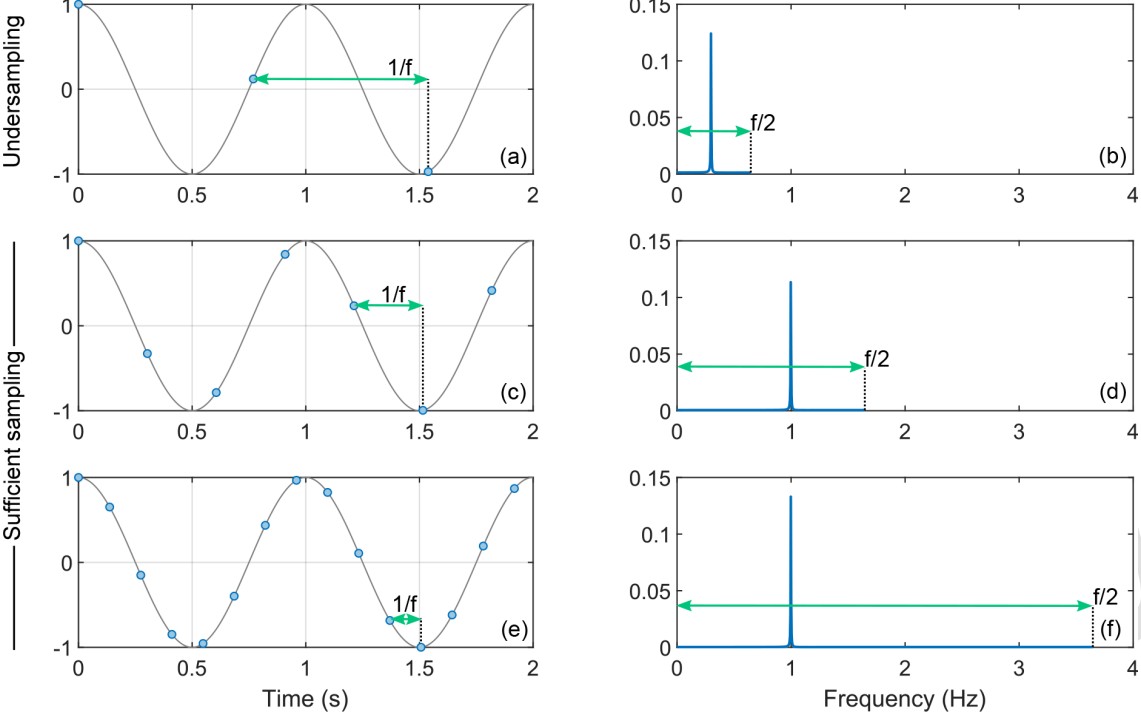

**Figure 3.** Illustration of signal sampling and the effect of undersampling. Panels (**a**), (**c**), and (**e**) TS10 represent a cosine oscillating at 1 Hz in gray sampled with various frequencies $f$ (1.3, 3.7, and 7.3 Hz). The blue dots represent the samples. In each case, the Fourier transform is shown in panels (**b**), (**d**), and (**f**). When the sampling frequency is lower than 1 Hz, the peak cannot appear at 1 Hz and is therefore found at a fictitious position.

and keeping other parameters constant. If the sampling frequency is sufficient, the spectrum should not be affected.

The choice of the parameters discussed in this section and above influences the outcome of the simulation in the same way as it does for the experiment. Once an NMR simulation is running, one might want to play with the combination of $f$, $K$, $t_{aq}$, $T_2$, and $L$ until the simulated spectra display convenient features. If one intends to simulate spectra to match experimental data, one might simply perform the simulation with the same $f$, $K$, $L$, and $t_{aq}$ values. Table 1 summarizes the parameters which were used in this paper TS11.

The procedure described here yields an NMR signal which is symmetric around 0. As a consequence, each signal is found both in the positive and negative frequencies, and the integral is split into the two duplicates. Because the experimental procedure that we are simulating does not differentiate negative and positive frequencies, we discard the frequency domain signal corresponding to negative frequencies and multiply the abscissa of the frequency domain signal corresponding to positive frequencies by a factor of 2. This operation corresponds to "folding" the spectrum around $\nu = 0$. Note that in high-field NMR, the measured signal is complex and is therefore not split into positive and negative halves. The central frequency of the spectrum at high field is given by the carrier frequency of the spectrometer (e.g., typically

400 MHz for $^1$H at 9.4 T). Section 2.8 describe this difference between high-field and ZULF NMR in more detail.

Whether the time domain signal which results from the simulation is real or complex, the Fourier transform yields a complex frequency domain signal. To get a spectrum consisting of a signal intensity as a function of the frequency, we must use the real part of the frequency domain signal. Depending on the experiment that we are simulating, we might find that some or all spectral components of the frequency domain signal are not in phase. To compensate for this, one might apply a phase correction by multiplying each point of the frequency domain signal by a complex constant $\exp i\phi$, where $\varphi$ is the phase correction before taking its real part.

$$I_r(\nu) = Re\{I_c(\nu)\exp i\phi\}, \tag{41}$$

where $I_r(\nu)$ and $I_c(\nu)$ are the real and complex frequency domain signals, respectively.

In summary, the Fourier transform procedure that we have described has the following steps:

1. Apply a monoexponential apodization window to the time domain signal so that it decays to 0 (see Eq. 36).

2. Apply the apodization described by Eq. (37) to avoid baseline artifacts in the frequency domain signal.

**Table 1.** List of parameters that were used to simulate the time domain signals and spectra in Fig. 4.

| Parameter | Meaning | Value used in Fig. 4 |
|---|---|---|
| $K$ | Number of points of the time domain signal | 4096 |
| $L$ | Number of points of the time domain signal including zero-filling/number of points of the Fourier transform | 65 536 |
| $t_{aq}$ | Acquisition time | 5 s |
| $f$ | Sampling frequency or spectral width | 819.200 Hz |
| $\tau_d$ | Dwell time (time between acquisition points) | 1.2207 ms |
| $T_2$ | Coherence's relaxation time constant | 1 s |
| $l_b$ | Line broadening | 0.3183 Hz |

3. Obtain the complex frequency domain signal by Fourier transforming the time domain signal using a fast Fourier transform algorithm.

4. Generate the corresponding frequency axis using Eqs. (39) and (40).

5. Remove the negative frequencies from both the frequency axis and the frequency domain signal and multiply the abscissa of the frequency domain signal by 2 to account for the partition of the signal integral between positive and negative frequencies.

6. Take the real part of the signal.

## 2.8   Comparison with high-field NMR

We conclude this theory section by listing the main differences between high-field and ZULF NMR, which are summarized in Table 2. As is the case for the rest of the paper, our description is limited to molecules containing spin-1/2 in the liquid state.

At high magnetic field, the Zeeman interaction dominates the dynamics and the $J$ coupling. Furthermore, the Larmor frequency of the spins (which results from the Zeeman interaction) is slightly shifted by the presence of the electron cloud around the nuclei. This phenomenon, called the chemical shift, gives a slightly different Larmor frequency for nuclei in different positions in a molecule, which spreads over typically 10 and 200 ppm around the Larmor frequency for $^1$H and $^{13}$C spins, respectively. At ZULF, the $J$ coupling dominates while the Zeeman interaction is a perturbation, and the chemical shift plays no role (in that it is a small perturbation of a small perturbation).

In Fig. 1 and in the simulations presented in this paper, we have assumed that the detector was positioned below the sample (along the $z$ axis in our axis convention) and that it was sensitive to magnetic field along the $z$ axis. Although this choice is typical, it is not the only possibility. In common high-field experiments, the oscillating signal emitted by the spins is recorded perpendicular to the static magnetic field. Detection at ZULF is performed with magnetometers that are sensitive to the total magnetic field produced by the sample. The operator corresponding to this observable is the sum of the magnetic moment of the spins along the sensitive axis of the OPM (see Eq. 34). In typical experiments, a single detector is used, which results in a real signal. Note that an imaginary ZULF signal could be obtained if the OPM were to have several sensitive axes or more than one detector were used. High-field NMR uses Faraday induction in pickup coils. Signals originating from different nuclei are usually not observed in the same experiment as their Larmor frequencies are too far apart, and the NMR coils are only sensitive over a limited bandwidth. The operator corresponding to inductive detection in pulsed NMR is non-Hermitean and therefore yields complex signals. An extra step of the acquisition process at high field that is not required at ZULF is modulating the signal recorded by the coil with a carrier frequency. Indeed, the NMR coil picks up a signal at the Larmor frequency of the spins, which is too high to be digitized (e.g., 400 MHz for $^1$H at 9.4 T). Instead, the signal is mixed with a carrier frequency, and only the difference is digitized, over a small bandwidth (e.g., over 10 ppm, corresponding to 4 kHz for $^1$H spins at 9.4 T). The signals at ZULF can be detected without mixing the frequency as the they are sufficiently low to be digitalized directly. For more details on the signal modulation at high field, the reader is referred to chapter 4 of J. Keeler's TS12 book *Understanding NMR spectroscopy* (Keeler, 2010).

The code in Sect. S2.2 TS13 presents in great detail the simulation of the spectra for a pair of $J$-coupled $^1$H and $^{13}$C spin pairs at ZF and ULF and at high field for both $^1$H and $^{13}$C (9.4 T). The code is decomposed in sections corresponding to Sect. 2.1 to 2.7 of the text above, and, whenever possible, the equations presented in this paper are referenced in the code. The reader is encouraged to open this code to understand the difference between simulating a spectrum at high field and at ZULF. The code can be opened in PDF, including the figures, for those who do not have a MATLAB license.

**Table 2.** Comparison between high-field and ZULF NMR for typical experiments. Note that quadrature detection (and thus imaginary signals) is possible at ZULF, although uncommon.

|  | ZULF | High field |
| --- | --- | --- |
| Main interaction | $J$-coupling $\hat{H}_J$ | Zeeman interaction $\hat{H}_Z$ |
| Perturbations | Zeeman interaction $\hat{H}_Z$ | $J$-coupling $\hat{H}_J$, chemical shift $\hat{H}_{CS}$ |
| Detection method | Magnetometry (OPM, SQUID CE5, etc.) | Faraday induction |
| Observables | $\hat{\mu}_{S,z} + \hat{\mu}_{I,z} = \gamma_I \hat{I}_z + \gamma_S \hat{S}_z$ | $\hat{I}_z = \hat{I}_x i \hat{I}_y$ |
| Signal type | Real | Complex |

## 3 Results of numerical simulations

### 3.1 Excitation schemes on an XA spin system

The ZF and ULF spectra of an XA spin system with a $J$ coupling of 140 Hz were simulated for different experimental sequences, assuming that the sample consists of 100 µL of solution where the spin system has a concentration of 27 mol L$^{-1}$. The code and its PDF version are presented in Sect. S2.3 TS14. Figure 4 shows the experimental sequences, as well as the simulated time domain and frequency domain signals. For all simulations, the sample was assumed to have spent sufficient time in a prepolarizing field of 2 T at 298 K to be at Boltzmann's equilibrium. The polarizations of the $^{13}$C and $^1$H spins were calculated using Boltzmann's distribution (see Eq. 19) and used to compute the single-spin density matrices of the $^{13}$C and $^1$H spins, $\hat{\rho}_{eq}(^{13}\text{C})$ and $\hat{\rho}_{eq}(^1\text{H})$ (see Eq. 21). The density matrix of the two-spin system was computed by taking the Kronecker product of the single-spin density matrices $\hat{\rho}_0 = \hat{\rho}_{eq}(^{13}\text{C}) \otimes \hat{\rho}_{eq}(^1\text{H})$ (see Eq. 23). The identity was removed from the two-spin density matrix using Eq. (24). The resulting density matrix was assumed to represent the initial state of the simulation (as explained above, only the Zeeman terms are considered to contribute to the initial state). For each experimental sequence, the spectrum was simulated both at 0 nT (including only the $J$ Hamiltonian $\hat{H}_J$; see Eq. 12) and with a field of 0.5 µT along the $x$ axis, that is, orthogonal to both the direction of the prepolarizing field and the sensitive axis (including both the $J$ Hamiltonian $\hat{H}_J$ and the Zeeman Hamiltonian $\hat{H}_Z$; see Eqs. 12 and 10). The time domain signal was computed by propagating the density matrix under the effect of the Hamiltonian for a total time of 5 s (parameter $t_{aq}$), discretized into 4096 points (parameter $K$), and corresponding to time intervals d$t$ of 1.2207 ms (parameter $\tau_D$). Prior to the propagation loop, the ZF and ULF propagators for this particular time step $\hat{U}$ (see Eq. 30) and the observable operator $\hat{O}$ (see Eq. 35) were computed only once.

The density matrix was propagated from time $t_k$ to time $t_{k+1} = t_k + \text{d}t$ under the Hamiltonian (ZF or ULF) using the sandwich formula $\hat{\rho}_{k+1} = \hat{U} \hat{\rho}_k \hat{U}^{-1} = \hat{U} \hat{\rho}_k \hat{U}^\dagger$ (see Eq. 29). At each time point $k$ of the propagation (realized by a "for" loop), the signal intensity of the time domain signal was extracted from the density matrix using the trace $\text{Tr}\left\{\hat{O}\hat{\rho}_k\right\}$ (see Eq. 33) (in pT). In theory, the trace of a Hermitean operator should be real. However, due to the finite machine precision of the numeric algorithm, the trace can sometimes contain a nonzero imaginary part. This residual imaginary part is discarded by taking the real part of the trace $Re\left(\text{Tr}\left\{\hat{O}\hat{\rho}_k\right\}\right)$. This point might appear secondary, but dealing with complex numbers while thinking they are real can lead to mistakes. After propagation, a monoexponential apodization function was applied to the time domain signal (see Eq. 36), with a coherence time constant $T_2$ of 1 s. A second apodization function was applied to avoid baseline artifacts (see Eq. 37). The apodized time domain signal was Fourier transformed with zero-filling to 65 536 points, using MATLAB's built-in functions. The real part of the Fourier transform is shown in Fig. 4. The frequency axis of the spectra was computed using Eqs. (39) and (40). The spectra are symmetric around zero, and so it is common to work only with the positive frequencies as shown in Fig. 4.

Simulating the sudden field drop experiment is the simplest case presented here. Because the coherence excitation scheme (or mixing) only consists of bringing the spin from high magnetic field to ZF or ULF, the simulation only consists of propagating the high-field thermal equilibrium density matrix under the ZF or ULF Hamiltonian. The ZF spectrum consists of one line at the $J$ coupling and one at zero frequency (see Fig. 4a). Including a field of 0.5 µT along the $x$ axis (ULF case) splits the $J$ peak as well as the line at zero frequency.

The simulations presented in Fig. 4b–d feature an adiabatic field drop. We used a monoexponential field drop from $B_{start} = 200$ µT to 0 occurring over $t_{decay} = 0.5$ s with a decay time constant of $\tau = 0.05$ s, described by

$$B(t) = B_{start} \frac{\exp\left(-\frac{t}{\tau}\right) - \exp\left(-\frac{t_{decay}}{\tau}\right)}{1 - \exp\left(-\frac{t_{decay}}{\tau}\right)}, \quad (42)$$

which fulfills the conditions $B(0) = B_{start}$ and $B(t_{decay}) = 0$. During the field drop, the Hamiltonian $\hat{H}(t) = \hat{H}_J + \hat{H}_Z(t)$ is time dependent. This step thus cannot be simulated in a single propagation step. Instead, it must be discretized into substeps d$t$ that are sufficiently short for the Hamilto-

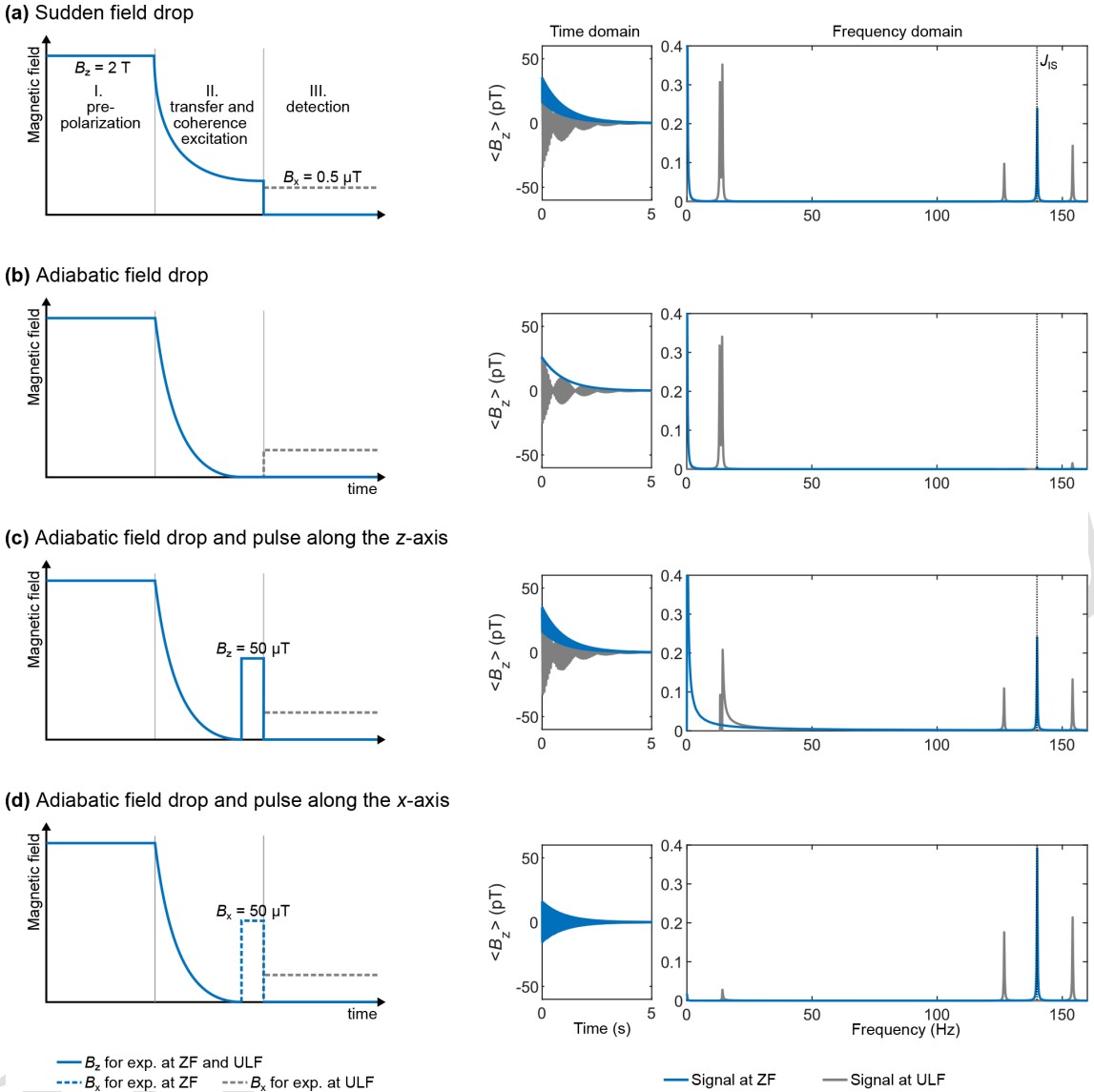

**Figure 4.** Excitation schemes for an XA spin system corresponding to $^{13}$C and $^{1}$H spins with a $J$ coupling of 140 Hz and corresponding simulated time domain signals and spectra. The vertical dashed line indicates the $J$ coupling. The time domain signal was computed by propagating the density matrix under the effect of the Hamiltonian for a total time of 5 s (parameter $t_{aq}$), discretized into 4096 points (parameter $K$), and corresponding to time intervals d$t$ of 1.2207 ms (parameter $\tau_D$). A monoexponential apodization function was applied to the time domain signal, with a coherence time constant $T_2$ of 1 s. The apodized time domain signal was Fourier transformed with a zero-filling to 65 536 points.

nian to be considered time independent. Here, the 0.5 s time length was discretized into 5000 steps of 0.1 ms. At time $t = 0$, the density matrix is the thermal equilibrium density matrix $\hat{\rho}_{eq}$ obtained above. At each time step $t_k$, the prop-
agator $\hat{U}(t_k \rightarrow t_{k+1}) = \exp\left(-i\hat{H}(t_k)dt\right)$ is computed (see Eq. 30), and the density matrix is propagated from time $t_k$ to time $t_{k+1} = t_k + dt$ (see Eq. 29) under the Hamiltonian $\hat{H}(t_k) = \hat{H}_J + \hat{H}_Z(t_k)$. We name $\hat{\rho}_{adia}$ the density matrix obtained after this process. A question arises here: is this magnetic field drop that we have chosen sufficiently slow to be considered adiabatic? In other words, is $\hat{\rho}_{adia}$ stationary? A simple way to ensure that it is the case is to simulate the spectrum at ZF after the magnetic field drop without any excitation pulse, that is, taking $\hat{\rho}_{adia}$ as the density matrix at time $t = 0$, $\hat{\rho}_0$. If the transition is adiabatic, then the system should remain stationary; that is, the time domain signal should feature no oscillation and the spectrum no peak. Figure 4b shows the result of this procedure, which confirms that the transition is adiabatic. The only feature of the ZF spectrum in Fig. 4b is the line at zero frequency. This line originates from the

non-oscillating magnetization decaying with $T_2$, which is the result of the apodization function that we have applied. Verifying that the ZF spectrum is flat also ensures that the field drop was discretized into sufficiently short time intervals d$t$.

The density matrix after the adiabatic field drop $\hat{\rho}_{\text{adia}}$ obtained above was used for the simulations presented in Fig. 4c–d. In the experimental sequences of Fig. 4c–d, the adiabatic field drop is followed by a magnetic field pulse either along the $z$ or $x$ axis. This was simulated by propagating $\hat{\rho}_{\text{adia}}$ under the pulse Hamiltonian to obtain $\hat{\rho}_0 = \hat{U}_p(\tau_p)\hat{\rho}_{\text{adia}}\hat{U}_p^\dagger(\tau_p)$, where $\hat{U}_p(\tau_p)$ is the propagator of the pulse Hamiltonian $\hat{H}_p = \hat{H}_J + \hat{H}_Z$, which acts on the density matrix during pulse length $\tau_p$. The Zeeman Hamiltonian depends on the magnetic field intensity of the pulse $B_p$ and its direction (see Eq. 15). For the $z$-axis pulse, we used a pulse intensity and length of $50\,\mu$T and $150\,\mu$s, respectively. For the $x$-axis pulse, we used a pulse intensity and length of $50\,\mu$T and $910\,\mu$s, respectively. These choices are justified in the next section. The resulting density matrices $\hat{\rho}_0$ were used as the density matrix at time $t = 0$ of the time domain signal, which was computed and Fourier transformed as described above. In the case of the $z$-axis pulse experiment, the peaks of interest ($J$ peak at 140 Hz) were found to be out of phase; a phase correction $e^{i\phi}$ with $\phi = \pi/2$ was thus applied to the Fourier transform. Adjusting the phase for the $J$ peak caused the lower-frequency peaks to be out of phase. Interestingly, in Fig. 4d, the intensity of the $J$ peak is higher than for the other excitation schemes while the lower-frequency peaks are suppressed, indicating that all the available polarization has been transferred to the $J$ peak.

## 3.2 Rabi oscillation curves

The pairs of magnetic field intensity and length of the pulses used for the simulation in Fig. 4d were chosen by simulating Rabi curves for both the $z$- and $x$-axis pulses. The high-field NMR equivalent to the Rabi curve is the "nutation experiment", which consists of recording a series of NMR detections while keeping the RF pulse power constant and varying the pulse length (or the pulse length is kept constant and the amplitude is varied; Tayler et al., 2017). The nutation or Rabi curve is the plot of the signal intensity as a function of the varied parameter. It allows us to determine the pair of RF power and pulse length which maximizes the signal intensity. Except in the presence of rapid relaxation effects or RF field inhomogeneities, the observed curve is sinusoidal. At ZULF, the Rabi curve is more complex and depends on the spin system under scrutiny. To simulate the Rabi curve at ZF, we repeated the simulation of the ZF spectra for an experiment with an adiabatic field drop (using the same parameters as above) followed by a pulse of $50\,\mu$T along the $z$ and $x$ axes, varying the pulse length from 0 to $3000\,\mu$s (the code and its PDF version are presented in Sect. S2.3 TS15). The time domain signal was Fourier transformed as described above, and the frequency domain signal was integrated from

138 to 142 Hz. The signal integral of the $J$ peak is plotted as a function of the pulse length in Fig. 5. The signal integral of the sudden drop experiment is shown as a horizontal dashed line for comparison. When a pulse along the $z$ axis is used, a simple sinusoidal curve is obtained, and its maximum matches that of the sudden drop experiment (see Fig. 5a). The first maximum is reached for a pulse length of $150\,\mu$s. When a pulse along the $x$ axis is used, a more complex pattern is obtained, and the maximum is found to be 1.64 times higher than the sudden drop experiment (see Fig. 5b). The first global maximum is reached for a pulse length of $910\,\mu$s.

## 3.3 X$A_n$ spin system

The simulations shown up to this point only deal with an XA spin system, which typically corresponds to $^{13}$C-formate (or $^{13}$C-formic acid), where the $^{13}$C spin interacts with a single $^1$H through a $J$ coupling of 195–222 Hz (Blanchard and Budker, 2016; Tayler et al., 2017) (depending on experimental conditions). $^{13}$C,$^{15}$N-cyanide groups are also interesting two-spin systems which were used in ZULF experiments (Blanchard et al., 2020, 2015). We now extend the simulation to incorporate multiple A spins. An X$A_2$ spin system is, for example, met in $^{13}$C-glycine (Put et al., 2021). X$A_3$ spins are met in a number of molecules containing methyl groups such as $^{13}$C-pyruvate (Barskiy et al., 2019). X$A_4$ (for example $^{15}$N-ammonium cation; Barskiy et al., 2019) and X$A_5$ are less common, but they are presented here to show the pattern that arises when adding spins.

Figure 6 shows the simulations for sudden drop experiments with detection at ZF and ULF of X$A_n$ spin systems with $n = 1, 2, \ldots, 5$, where X represents a $^{13}$C spin, and $A_n$ represents $^1$H spins with a $J$ coupling of 140 Hz between X and A spins and 10 Hz among A spins (the code and its PDF version are presented in Sect. S2.4 TS16). All the relevant mathematics to construct the operators of an $m = n + 1$ spin system are given in the Theory section. For an X$A_5$ spin system, the Hilbert space has $2^6 = 64$ dimensions (and related operators). To avoid constructing each operator manually, recursive formulae were used (see Eqs. 13 and 23). The time domain signal was computed by propagating the density matrix under the effect of the Hamiltonian for a total time of 5 s (parameter $t_{\text{aq}}$), discretized into 8192 points (parameter $K$), and corresponding to time intervals d$t$ of 0.6104 ms (parameter $\tau_D$). A monoexponential apodization function was applied to the time domain signal, with a coherence time constant $T_2$ of 1 s. The apodized time domain signal was Fourier transformed with a zero-filling to 32 768 points.

Increasing the number of A spins increases the number of spectral components in the spectrum. A known result of ZULF NMR appears in this simulation: for odd numbers of $n$, the ZF spectrum features lines at integer multiples of the $J$ coupling $k \cdot J_{\text{AX}}$ TS17 with $k \in [\![1; (n+1)/2]\!]$, while for even numbers of $n$, it features lines at half-integer multiples of the $J$ coupling $k \cdot J_{\text{AX}}/2$ with $k \in [\![1; n/2]\!]$. Adding a $0.5\,\mu$T

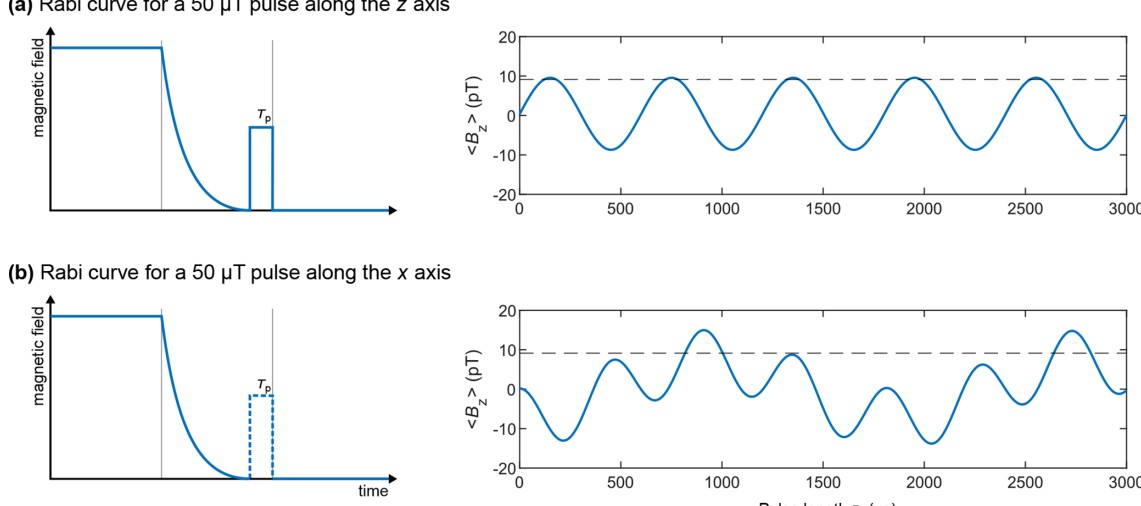

**Figure 5.** Rabi curves at ZF with excitation pulses along $z$ **(a)** and $x$ **(b)** axes applied to an XA spin system. The horizontal dashed line represents the signal integral of the sudden drop ZF experiment. The time domain signal was computed by propagating the density matrix under the effect of the Hamiltonian for a total time of 5 s (parameter $t_{aq}$), discretized into 4096 points (parameter $K$), and corresponding to time intervals d$t$ of 1.2207 ms (parameter $\tau_D$). A monoexponential apodization function was applied to the time domain signal, with a coherence time constant $T_2$ of 1 s. The apodized time domain signal was Fourier transformed with a zero-filling to 65 536 points. The frequency domain signal was then integrated from 138 to 142 Hz. The Rabi curve represents the integral compared with the excitation pulse length. The picotesla unit (pT) CE6 should be recalculated.

field along the $x$ axis during detection (that is, performing ULF detection) splits the $J$ lines. The higher the multiple of the $J$ line, the greater the number of splittings. Note that the intensity of NMR signals at high field increases upon
adding more equivalent spins to the spin system. The analysis of Fig. 6 shows that this logic does not apply to the $J$ lines for the ZULF case, where the spectrum completely changes upon changing of spin topology. For example, note that the amplitude of the $J$ line for the XA system have the
same intensity as the $J$ line for the XA$_2$ system (appearing at $3/2 \cdot J_{AX}$ frequency). Likewise, the two $J$ lines for the XA$_3$ system has the same total intensity as the two $J$ lines for the XA$_4$ system. An empirical law of conservation of the total spectral intensity for the $J$ lines can be deduced by looking at
Fig. 6: indeed, the total intensity of all $J$ lines is the same for any XA$_n$ system, assuming equal sample volume, prepolarization, etc. On the other hand, the intensity of low-frequency peaks shown in Fig. 6 is proportional to the total number of spins in the spin system, like in high-field NMR. This is of
course expected as these signals are associated with the precession of total magnetization around residual ULF field, and total magnetization is proportional to the number of spins.

## 4 Interpretation

We are now going to show how to calculate ZULF NMR
spectra considering energy levels and transition probabilities rather than through the numerical propagation of the density matrix. We will derive analytical solutions for the XA$_n$ sys-

tem, but the same approach can be used for more complex spin systems. This approach was investigated in the following references: Butler et al., 2013a; Theis et al., 2013; and
30 Emondts et al., 2014. Here we aim to present it with more explanations and explicit derivations, but we limit ourselves to only the simplest spin systems.

The relative contribution of $\hat{H}_Z$ (see Eq. 10) and $\hat{H}_J$ (see Eq. 12) terms depends on the magnetic field strength. In the
35 high-field extreme, for a heteronuclear spin system, $\hat{H}_Z$ is the dominant term, and $\hat{H}_J$ is considered as a first-order perturbation. In this case, heteronuclei are said to be weakly coupled, and their eigenstates coincide with the Zeeman states (e.g., those in Eq. 9). At zero field, the weak coupling approx-
40 imation is not valid; the Zeeman states do not correspond to the eigenstates of system. However, it is still possible to calculate analytically the eigenstates for some spin systems, and the simplest case is when all the spins are identical (A$_n$ system). In this case, the Hamiltonian is represented by only the
45 $\hat{H}_J$ term, and it commutes with the square of the total angular momentum operator TS18.

$$\hat{F}^2 = \hat{F}_x^2 + \hat{F}_y^2 + \hat{F}_z^2,$$

$$\hat{F}_\mu = \sum_{l=1}^{n} \hat{I}_{l\mu}; \ \mu \in \{x, y, z\},$$

$$\left[\hat{H}_J, \hat{F}^2\right] = \hat{H}_J \hat{F}^2 - \hat{F}^2 \hat{H}_J = 0, \tag{43}$$

where $n$ is the number of spins in the system. It is well known that any pair of commuting Hermitean operators share

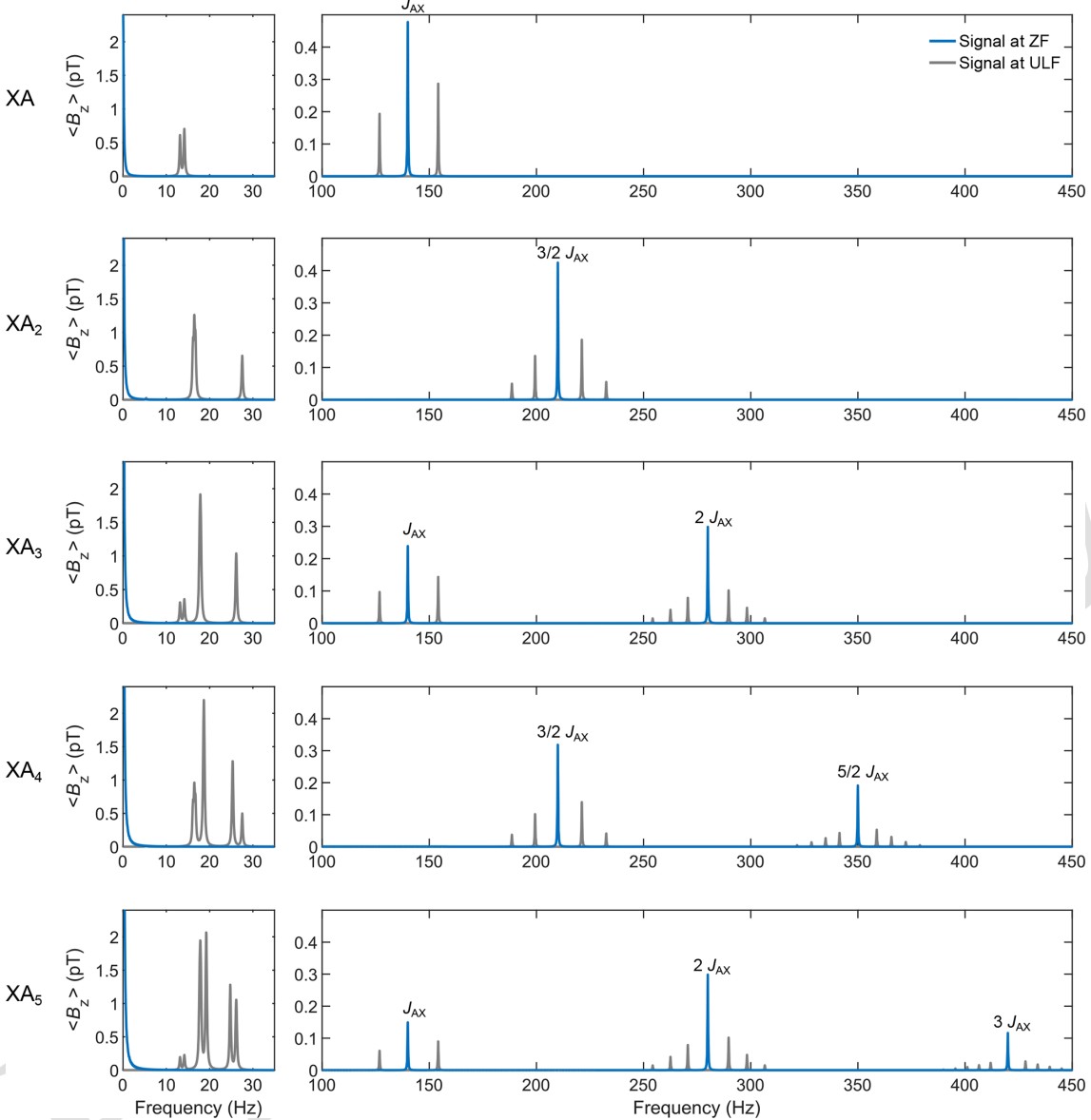

**Figure 6.** Simulation of ZF and ULF spectra after sudden field drop for XA, $XA_2$, $XA_3$, $XA_4$, and $XA_5$ spin systems with a $J$ coupling of 140 Hz between X and A spins and 10 Hz among A spins. The time domain signal was computed by propagating the density matrix under the effect of the Hamiltonian for a total time of 5 s (parameter $t_{aq}$), discretized into 4096 points (parameter $K$), and corresponding to time intervals d$t$ of 1.2207 ms (parameter $\tau_D$). A monoexponential apodization function was applied to the time domain signal, with a coherence time constant $T_2$ of 1 s. The apodized time domain signal was Fourier transformed with a zero-filling to 32 768 points.

their eigenspaces (Levitt, 2013). The set of eigenstates which forms an eigenbasis for both operators simultaneously is unique in cases where there are no degeneracies (all the eigenvalues for both operators are different). When there are degeneracies, the common eigenbasis is not unique. It turns out that $\hat{H}_J$ and $\hat{F}^2$ operators have degeneracies, and this results in the existence of an infinite number of different shared eigenbases. Let us describe how to find such a set of eigenstates.

## 4.1 Eigenstates at zero field

The eigenstates of a $\hat{F}^2$ operator can be expressed in terms of the total spin and its projection quantum numbers. The conventional way to express them is to use the $|F, m_F\rangle$ notation, where $F$ denotes the total spin, and $m_F$ denotes the projection onto a quantization axis ($m_F \in \{-F, -F+1, ..., F-1, F\}$). For example, by definition, for a single spin of 1/2, we have the sates $|\alpha\rangle \equiv |1/2, 1/2\rangle$ and $|\beta\rangle \equiv |1/2, -1/2\rangle$. For a pair of spins, we have the three triplet states $|T_{+1}\rangle \equiv |1, 1\rangle$; $|T_0\rangle \equiv$

**Figure 7.** Procedure for adding up the angular momenta for the $A_3$ spin system.

$|1, 0\rangle$; and $|T_{-1}\rangle \equiv |1, -1\rangle$; and the singlet state $|S_0\rangle \equiv |0, 0\rangle$. Any $|F, m_F\rangle$ state is an eigenstate of the $\hat{F}^2$ and $\hat{F}_z$ operators with the following eigenvalues: TS19

$$\hat{F}^2 |F, m_F\rangle = F(F+1) |F, m_F\rangle,$$

$$\hat{F}_z |F, m_F\rangle = m_F |F, m_F\rangle. \qquad (44)$$

To find the total spin of a system constituted by $n$ spins, one must sum up the angular momenta of the individual spins, which is a common procedure in the field of atomic physics but not so much in NMR. All possible values of the angular momentum of the interacting spins are added up to con-
stitute a set of uncoupled quasiparticles with different total spin. The total spin $F$ of a system constituted by two spins $I$ and $S$ can take the values with steps of 1 between the sum $I + S$ and the absolute value of their difference:

$$|I - S| \le F \le I + S. \qquad (45)$$

For a pair of spins of 1/2, the possible values are $F = 0, 1$. For $n$ spins, the summation should proceed until all the possible pairs of the angular momentum of the individual spins are summed up. As an illustration, consider a coupled system of three spins of 1/2 (see Fig. 7). First, any two spins are added
up together to give $F = 1$ (a triplet) and $F = 0$ (a singlet). Then, the remaining spin 1/2 is added up to the quasiparticles formed in the previous step (spins 1 and 0 in this case). As a result, the initial $A_3$ system is decomposed into three subsystems with total spins of $F = 3/2$, 1/2 (addition of 1
and 1/2), and 1/2 (addition of 0 and 1/2).

A useful property of such a decomposition can be illustrated at this point: the total spin operator commutes with all rotation operators (e.g., $\exp(-i\theta \hat{I}_z)$); therefore, 3D rotations will never mix terms of the wave function belonging to differ-
30 ent total spin, e.g., spin 3/2 with 1/2. At ZF, there is no distinction between directions; therefore, the eigenstates must be invariant with respect to 3D rotations. This also partially explains the existence of an infinite number of eigenbases for $\hat{F}^2$, as all different orientations of the $\{x, y, z\}$ system corre-
35 spond to different bases.

One can check that the total number of the spin states remains the same after the procedure of adding up the spins. On the one hand, the number of states formed by $n$ coupled spins $I$ equals to $(2I + 1)^n$, which is 8 in the considered case.
On the other hand, a manifold CE7 with a total spin $F$ has $2F + 1$ different states associated with different possible projections of the spin on the quantization axis. Therefore, there are $4 + 2 + 2$ states in the considered case.

The explicit form of the resulting eigenstates can be ob-
45 tained in terms of "uncoupled" spin states, which are constructed as a Kronecker product of the individual Zeeman states (see Eq. 9). The resulting state $|F, m_F\rangle$ of the addition of two angular momenta ($I$ and $S$) can be represented as the following linear combination:

$$|F, m_F\rangle = \sum_{m_I, m_S} C^{F, m_F}_{I, m_I, S, m_S} |I, m_I, S, m_S\rangle, \qquad (46)$$

where $C^{F, m_F}_{I, m_I, J, m_J}$ represents Clebsch–Gordan coefficients and is defined by

$$C^{F, m_F}_{I, m_I, S, m_S} = \langle I, m_I; S, m_S | F, m_F\rangle. \qquad (47)$$

Each Clebsch–Gordan coefficient is specified by six numbers: the total spin of the coupled state $F$, its projection $m_F$,
and the total spins of the uncoupled states and their projections ($I, S, m_I, m_S$). Coefficient $C^{F, m_F}_{I, m_I, S, m_S}$ represents "how much" of uncoupled state $|I, m_I, S, m_S\rangle$ there is in a coupled state $|F, m_F\rangle$. The analytical values of the Clebsch–
Gordan coefficients can be calculated using recursive expressions, which are available in many software packages and textbooks. Table S1 in the Supplement provides the relation between the coupled and uncoupled states for the considered $A_3$ system and shows explicitly how to calculate them. The
full set of all possible $|F, m_F\rangle$ states forms the new basis that is better suited than the Zeeman basis for ZULF NMR. In fact, this basis coincides with the eigenstates at ZULF for $A_n$ and for $XA_n$ systems, but this basis is also a good starting point for more complicated cases. We will refer to this new
basis as a "coupled" basis, because it is appropriate for the description of strongly coupled spins.

## 4.2 Eigenenergies at zero field

Having the eigenstates, we can now proceed with finding the eigenvalues of the Hamiltonian; these values correspond to the energy of the states and therefore determine the fre-
75 quencies of ZULF NMR transitions. It turns out that $A_n$ systems are not detectable at ZULF; it is shown in the next section (where intensities of transitions are calculated) that they give rise to no observable transition. At least two types of nuclei with different gyromagnetic ratios are necessary for
an observable transition to exist. Therefore, we consider an $XA_n$ system from now on. We will denote the operators associated with the X spin as $\hat{S}$ and with A spins as $\hat{I}$. It is also convenient to introduce total spin operators for A spins: $\hat{F}_{A\mu} = \sum_{l=1}^{n} \hat{I}_{l\mu}$, $\mu \in \{x, y, z\}$. The Hamiltonian at ZF for this
spin system is given by

$$\hat{H}_{AX} = 2\pi J_{AX} \sum_{l=1}^{n} \hat{S} \cdot \hat{I}_l + 2\pi J_{AA} \sum_{l=1}^{n-1} \sum_{k>l}^{n} \hat{I}_l \cdot \hat{I}_k. \qquad (48)$$

The $\hat{H}_{AX}$ Hamiltonian can be expressed in terms of the total spin operators using algebraic tricks. We find an expression

for the first term of Eq. (48) in terms of $\hat{F}^2$, $\hat{F}_A^2$, and $\hat{S}^2$ by developing $\hat{F}^2$: TS20

$$
\begin{aligned}
\hat{F}^2 &= \left( \hat{S} + \sum_{l=1}^n \hat{I}_l \right)^2 = \hat{S}^2 + 2\hat{S} \cdot \sum_{l=1}^n \hat{I}_l + \left( \sum_{l=1}^n \hat{I}_l \right)^2 \\
&= \hat{S}^2 + 2 \sum_{l=1}^n \hat{S} \cdot \hat{I}_l + \hat{F}_A^2 \Leftrightarrow \sum_{l=1}^n \hat{S} \cdot \hat{I}_l \\
&= \frac{1}{2} \left( \hat{F}^2 - \hat{S}^2 - \hat{F}_A^2 \right).
\end{aligned} \tag{49}
$$

Similarly, we find an expression for the second term of Eq. (48) in terms of $\hat{F}_A^2$ and $\hat{I}_l^2$ by developing $\hat{F}_A^2$:

$$
\begin{aligned}
\hat{F}_A^2 &= \left( \sum_{l=1}^n \hat{I}_l \right)^2 = \sum_{l=1}^n \hat{I}_l^2 + 2 \sum_{l=1}^{n-1} \sum_{k>l}^n \hat{I}_l \cdot \hat{I}_k \\
&\Leftrightarrow \sum_{l=1}^{n-1} \sum_{k>l}^n \hat{I}_l \cdot \hat{I}_k = \frac{1}{2} \left( \hat{F}_A^2 - \sum_{l=1}^n \hat{I}_l^2 \right).
\end{aligned} \tag{50}
$$

By substituting the results of Eqs. (49) and (50) into Eq. (48), we obtain a form of the Hamiltonian for which the energies will be more easily calculated:

$$
\begin{aligned}
\hat{H}_{AX} &= 2\pi J_{AX} \frac{1}{2} \left( \hat{F}^2 - \hat{S}^2 - \hat{F}_A^2 \right) \\
&+ 2\pi J_{AA} \frac{1}{2} \left( \hat{F}_A^2 - \sum_{l=1}^n \hat{I}_l^2 \right).
\end{aligned} \tag{51}
$$

The $\hat{H}_{AX}$ Hamiltonian commutes with the $\hat{F}^2$ operator; therefore, they share eigenstates $|F, m_F\rangle$. So, the eigenenergies can be written as the expectation values of $|F, m_F\rangle$ with respect to $\hat{H}_{AX}$:

$$
E_{F, m_F} = \langle F, m_F | \hat{H}_{AX} | F, m_F \rangle. \tag{52}
$$

To calculate explicitly the eigenvalues, we substitute the Hamiltonian of Eq. (51) into Eq. (52) and use the following properties: TS21

$$
\begin{aligned}
\hat{F}^2 | F, m_F \rangle &= F(F+1) | F, m_F \rangle, \\
\hat{S}^2 | F, m_F \rangle &= S(S+1) | F, m_F \rangle, \\
\hat{F}_A^2 | F, m_F \rangle &= F_A(F_A+1) | F, m_F \rangle, \\
\hat{I}_l^2 | F, m_F \rangle &= I_l(I_l+1) | F, m_F \rangle,
\end{aligned} \tag{53}
$$

to obtain the final expression for the energy of level $|F, m_F\rangle$.

$$
\begin{aligned}
E_{F, m_F} &= \frac{J_{AX}}{2} \left[ F(F+1) - S(S+1) - F_A(F_A+1) \right] \\
&+ \frac{J_{AA}}{2} \left[ F_A(F_A+1) - n I_l(I_l+1) \right],
\end{aligned} \tag{54}
$$

expressed in hertz. Here, quantum number $F$ corresponds to the total spin of the full XA$_n$ system, $S$ corresponds to the

spin of the nucleus X, $F_A$ is the total spin of the A$_n$ spins, and $I_l$ is the spin of individual nuclei A. The energy does not depend on the spin projection, resulting in degeneracy of all $2F + 1$ levels with equal $F$.

The spin number $S$ is the same for all eigenstates (e.g., it is $1/2$ for $^{13}$C); similarly, all spins $I_l$ are the same, and for $^1$H spins they are equal to $1/2$. The remaining two quantum numbers $F$ and $F_A$ can have different values depending on the state, therefore removing degeneracy between some of the levels. Figure 8 presents the energy levels of XA, XA$_2$, and XA$_3$ systems at ZF calculated using Eq. (54). Mathematica codes to perform these calculations are available in the Supplement (Sect. S3 TS22).

### 4.3 Selection rules

We have now found the eigenstates and their energies, but not all transitions between any pair of states are allowed. The last step is to find the transition intensities and thus get the analytical appearance for the ZF NMR spectrum of an XA$_n$ system. There are certain selection rules specifying which transitions are in principle possible and which are forbidden, like those in high-field NMR, where only single quantum transitions are allowed. A general expression for the transition intensity between any two eigenstates $|F, m_F\rangle$ and $|F', m'_F\rangle$ is given by

$$
Y = \langle F', m'_F | \hat{\rho}_0 | F, m_F \rangle \langle F', m'_F | \hat{O} | F, m_F \rangle. \tag{55}
$$

We will explicitly calculate the transition intensity for the sudden field drop experiment. In this case, both the initial state $\hat{\rho}_0$ and the observation operator $\hat{O}$ are proportional to $\gamma_I \hat{F}_{A,z} + \gamma_S \hat{S}_z$ (as a reminder, $\hat{F}_{A,z} = \sum_{l=1}^n \hat{I}_{l,z}$). Therefore, the transition intensity becomes

$$
Y = \left\langle F', m'_F | \gamma_I \hat{F}_{A,z} + \gamma_S \hat{S}_z | F, m_F \right\rangle^2. \tag{56}
$$

This expression is an example of Fermi's golden rule that is used to calculate a transition's amplitude in different problems in quantum mechanics. Similar expressions can be found for the high-field NMR. By expressing the coupled states $|F, m_F\rangle$ in terms of an uncoupled basis (see Eq. 46), we find that

$$
\begin{aligned}
&\left( \gamma_I \hat{F}_{A,z} + \gamma_S \hat{S}_z \right) | F, m_F \rangle \\
&= \left( \gamma_I \hat{F}_{A,z} + \gamma_S \hat{S}_z \right) \sum_{m_A, m_S} C_{F_A, m_A, S, m_S}^{F, m_F} | F_A, m_A, S, m_S \rangle \\
&= \sum_{m_A, m_S} C_{F_A, m_A, S, m_S}^{F, m_F} (\gamma_I m_A + \gamma_S m_S) | F_A, m_A, S, m_S \rangle,
\end{aligned} \tag{57}
$$

where $m_A$ and $m_S$ are the $z$ projection of the total spins $F_A$ (for $n$ protons, the maximum value of $F_A$ equals $n/2$, and for each value of $F_A$, $m_A \in \{-F_A, -F_A+1, \ldots, F_A-1, F_A\}$) and the $z$ projection of the spin $S$ (in the case where $S$ is

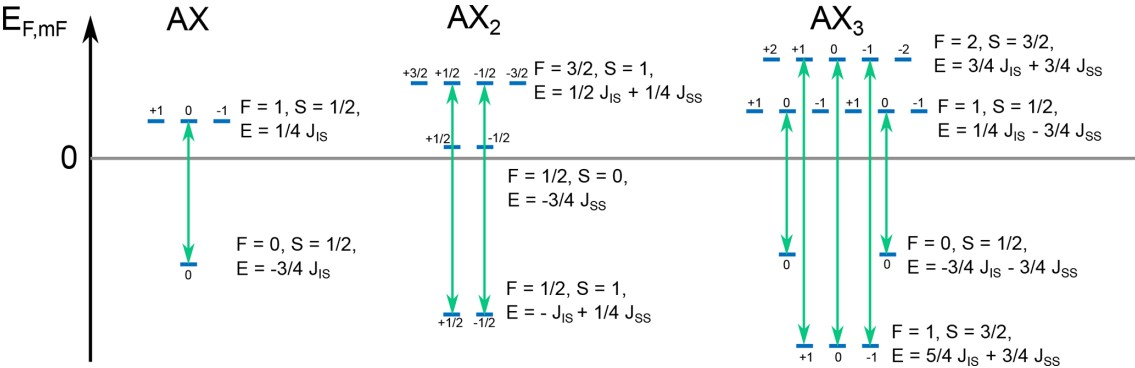

**Figure 8.** Energy levels for XA, XA$_2$, and XA$_3$ spin systems calculated according to Eq. (54). The numbers above the energy levels represent the $z$ projection of the angular momentum of the states $m_F$. Allowed transitions are shown by green arrows. $J_{AX}$ was set to 140 Hz, and $J_{AA}$ was set to $-12$ Hz; these are typical values for $1J_{CH}$ and $2J_{HH}$ $J$ couplings TS23. The energy difference for the allowed transitions equals to $J_{AX}$ for the XA system, $3/2J_{AX}$ for the XA$_2$ system, and two frequencies of $2J_{AX}$ and of $J_{AX}$ for the XA$_3$ system. This agrees with the numerical simulations shown in Fig. 6.

carbon-13, $m_S \in \{-\frac{1}{2}, \frac{1}{2}\}$), respectively. Now let us express the remaining $\langle F', m'_F |$ state in terms of an uncoupled basis as well and combine Eqs. (56) and (57) TS24.

$$Y = \left( \sum_{m'_A, m'_S} C^{F', m'_F}_{F'_A, m'_A, S', m'_S} \langle F'_A, m'_A, S, m'_S |  \right.$$

$$\sum_{m_A, m_S} C^{F, m_F}_{F_A, m_A, S, m_S} (\gamma_I m_A + \gamma_S m_S) \, | F_A, m_A, S, m_S \rangle \Big)^2$$

$$= \left( \sum_{m'_A, m'_S, m_A, m_S} C^{F', m'_F}_{F'_A, m'_A, S', m'_S} C^{F, m_F}_{F_A, m_A, S, m_S} \right.$$

$$(\gamma_I m_I + \gamma_S m_S) \langle F'_A, m'_A, S', m'_S | F_A, m_A, S, m_S \rangle \Big)^2 \quad (58)$$

The last term $\langle F'_A, m'_A, S', m'_S | F_A, m_A, S, m_S \rangle$ is nonzero only if TS25

$$\Delta F_A = F'_A - F_A = 0,$$
$$\Delta m_A = m'_A - m_A = 0,$$
$$\Delta S = S' - S = 0,$$
$$\Delta m_S = m'_S - m_S = 0. \quad (59)$$

These selection rules mean that the only allowed transitions are those which conserve the total spins $F_A$ and $S$ ($S$ is con-
10 served automatically as it can be only $1/2$, but $F_A$ can have different values), as well as their projections onto the reference axis. Equation (58) therefore simplifies to

$$Y = \left( \sum_{m_A, m_S} C^{F', m'_F}_{F_A, m_A, S, m_S} C^{F, m_F}_{F_A, m_A, S, m_S} (\gamma_I m_A + \gamma_S m_S) \right)^2. \quad (60)$$

It is important to notice that, in the case where $\gamma_I = \gamma_S$,
each element of this sum becomes zero. This is shown in the Wolfram Mathematica code for all observable transitions in XA, XA$_2$, and XA$_3$ systems and can be rationalized in general case by the following (see Sect. S3.1 TS26). The operator

$\gamma_I \hat{F}_{A,z} + \gamma_S \hat{S}_z$ (which is proportional to the initial state $\hat{\rho}_0$) can be rewritten as $\gamma_I \left( \hat{F}_{A,z} + \hat{S}_z \right) + (\gamma_S - \gamma_I) \hat{S}_z$. The first
term in this expression commutes with the $\hat{H}_{AX}$ Hamiltonian (see Eq. 48); therefore, it does not produce any observable coherences, whereas the second term does not commute with the $\hat{H}_{AX}$ and leads to ZULF signals.

Finally, there are two more selection rules that are derived
by implementing the Wigner–Eckart theorem. The considered case is equivalent to a "dipole" transition, where the transition is observed between two states connected by operator of rank 1 (e.g., Eq. 56). This is a common situation in atomic physics, and we adapt this result without evaluation:
the reduced matrix element coming from Wigner–Eckart is shown to be nonzero if and only if

$$\Delta F = \pm 1,$$
$$\Delta m_F = 0. \quad (61)$$

The whole set of selection rules given by Eqs. (59) and (61) allows us to find which transitions are observable in XA$_n$
systems at ZF. These transitions are shown in Fig. 8 by the green arrows. It can be seen that $J_{AA}$ couplings shift the energy levels but do not affect the frequencies of the observable transitions. This is a common situation that $J$ couplings between magnetically equivalent spins do not contribute to the
observed NMR spectrum. As can be seen from the analysis presented above, this statement holds for each case of the ZF NMR spectra of XA$_n$ systems.

In this section, we analytically found the allowed transitions for XA, XA$_2$, and XA$_3$ for the case of a sudden field
drop to ZF. The XA spin system has a single transition at $J_{AX}$, the XA$_2$ spin system has a single transition at $3/2J_{AX}$, and the XA$_3$ spin system has one allowed transition at $J_{AX}$ and another one at $2J_{AX}$. Allowed transitions analytically found here correspond to the numerical simulation: XA sin-
gle line at $J_{AX}$, XA$_2$ single line at $3/2J_{AX}$, etc. This deriva-

tion explained the appearance of the ZF spectra but not that of the ultralow-field spectra. To understand how the degeneracy of the ZF eigenstates are split by the presence of a bias field CE8, one has to use perturbation theory. We refer the interested reader to Ledbetter et al. (2011) and Appelt et al. (2010).

## 4.4 Rabi oscillation curves

We finish this section on the interpretation of the numerical simulations by giving a short explanation of the Rabi oscillation curves presented in Sect. 3.2 (see Fig. 5). The successful implementation of excitation pulses in ZULF-NMR experiments requires two conditions to be fulfilled (Butler et al., 2013b). First, the DC CE9 field of the pulse should be strong enough so that heteronuclei (here, $^1$H and $^{13}$C spins) can be considered weakly coupled. The field of $50\,\mu$T satisfies this condition, as the difference in Larmor frequencies between $^1$H and $^{13}$C spins is larger than $1.5\,\text{kHz} \gg J_{\text{XA}} = 140\,\text{Hz}$. Second, the pulse must be much shorter than the evolution under the $J$ coupling. Here, the longest pulses that were simulated had a duration of 3 ms, while the characteristic time of the evolution under the $J$ coupling is $1/J_{\text{XA}} \approx 7.1\,\text{ms}$. Provided these two conditions are met, the product operator formalism can provide a convenient explanation for the results of Fig. 5. Both Rabi oscillation curves in Fig. 5 are rather unusual for high-field NMR, but the reader who is familiar with the product operator formalism at high field will see that there is a strong connection between the algebra describing pulsed experiments at high field and at ZULF. Here, we give a brief summary of how this formalism can be used to understand the Rabi oscillation curves. We recommend the interested reader to look at the following references for a more detailed derivation (Butler et al., 2013b; Blanchard, 2014; Tayler et al., 2017).

After the adiabatic field drop, the magnetization of the sample is proportional to $\gamma_{\text{H}}\hat{I}_z + \gamma_{\text{C}}\hat{S}_z$ and does not evolve. In addition, part of the population is also on the zero-quantum term $\hat{Z}_z = 2\left(\hat{I}_x\hat{S}_x + \hat{I}_y\hat{S}_y\right)$, which produces no observable magnetization. Magnetic field pulses are applied to convert one or both of these terms into the observable zero-quantum term $\hat{Z}_x = \hat{I}_z - \hat{S}_z$. In the case of Fig. 5a where the pulse is applied along the $z$ axis, the pulse converts $\hat{Z}_z$ into $\hat{Z}_y = 2\left(\hat{I}_x\hat{S}_y + \hat{I}_y\hat{S}_x\right)$. The state of the system after the pulse is $\hat{\rho}\left(\tau_{\text{p}}\right) = \hat{Z}_z\cos\left[\left(\gamma_{\text{H}} - \gamma_{\text{C}}\right)B_z\tau_{\text{p}}\right] + \hat{Z}_y\sin\left[\left(\gamma_{\text{H}} - \gamma_{\text{C}}\right)B_z\tau_{\text{p}}\right]$. The excited unobservable $\hat{Z}_y$ term then starts to evolve into the observable term $\hat{Z}_x$ under the action of the $\hat{H}_J$ Hamiltonian, generating an oscillating magnetic field along the $z$ axis. The resulting ZULF signal has a sine rather than a cosine time dependence and requires a $\pi/2$ phase correction of the spectrum to have an absorption line as was described in Sect. 3.2. The signal is maximized when the pulse has duration $\tau_{\text{p}} = \pi/\left[2\left(\gamma_{\text{H}} - \gamma_{\text{C}}\right)B_z\right]$, which is around $157\,\mu$s in

the considered case. This is consistent with the simulated Rabi oscillation curve of Fig. 5a. In the case of Fig. 5b where the pulse is applied along the $x$ axis, both initial terms of the density operator, $\gamma_{\text{H}}\hat{I}_z + \gamma_{\text{C}}\hat{S}_z$ and $\hat{Z}_z$, are converted into the observable term $\hat{Z}_x$. The conversion follows a $\sin\left[\left(\gamma_{\text{H}} + \gamma_{\text{C}}\right)B_x\tau_{\text{p}}/2\right]\sin\left[\left(\gamma_{\text{H}} - \gamma_{\text{C}}\right)B_x\tau_{\text{p}}/2\right]$ function, allowing one to excite slightly stronger signals over a slightly longer pulse duration.

## 5 Conclusion

We have shown how to numerically simulate spectra at both zero and ultralow fields for sudden drop and pulsed experiments. We have then explained the results of the numerical simulation for sudden drop experiments at ZF by constructing the eigenbasis of the ZF Hamiltonian and finding the allowed transitions among the eigenstates. The other numerically simulated cases (i.e., pulsed experiments) can be explained using the analytical approach that we have presented here. It requires an additional step which is to describe how a pulse converts the populations of states. The reader who is acquainted with the product operator formalism commonly used in high-field NMR might be interested in an alternative approach based of commutation rules as presented in Blanchard and Budker (2016) and Butler et al. (2013b). We have chosen to describe the simplest cases, i.e., experiments with thermal prepolarization with AX$_n$ systems. Using this methodology, the reader can proceed with simulating more advanced cases, where analytical solutions do not exist. This includes calculation of ZULF spectra of molecules with multiple spins (Wilzewski et al., 2017) and molecules containing three or more types of nuclei, e.g., $^1$H, $^{13}$C, $^{15}$N, and $^2$D (Alcicek et al., 2021); the evolution during dynamical decoupling sequences (Bodenstedt et al., 2022a); the ZULF-TOCSY type of spin-locking experiments (Kiryutin et al., 2021); or spin evolution at intermediate fields, where perturbation approaches are not valid (Bodenstedt et al., 2021) CE10. Complicated spin dynamics may occur under the action of allowing composite pulses CE11 (Jiang et al., 2018; Bodenstedt et al., 2022b). The formalism we presented here is a good starting point for the description and understanding of hyperpolarized ZULF experiments, e.g., those involving transfer of spin order in parahydrogen experiments at low fields. Simulations are also useful to study different kinds of imperfections such as field inhomogeneities, timing errors, etc. We hope that this tutorial paper has allowed us to share our excitement with the reader.

**Code availability.** The codes used to simulate the spectra presented in this paper are available online (https://doi.org/10.5281/zenodo.7664138; Stern and Sheberstov, 2023). PDF versions of the codes are available in the Supplement.

**Data availability.** .TS27

**Supplement.** The supplement related to this article is available online at: https://doi.org/10.5194/mr-4-1-2023-supplement.

**Author contributions.** .TS28 .

**Competing interests.** The contact author has declared that neither of the authors has any competing interests.

**Disclaimer.** Publisher's note: Copernicus Publications remains neutral with regard to jurisdictional claims in published maps and institutional affiliations.

**Acknowledgements.** The Authors wish to thank Alice Stern for the drawing used in the key figure. This research was supported by ENS Lyon, the French CNRS, Lyon 1 University, the European Research Council under the European Union's Horizon 2020 research and innovation program (Marie Skłodowska-Curie grant agreement no. 766402/ZULF), and the French National Research Agency (project "HyMag" ANR-18-CE09-0013).

**Financial support.** This research has been supported by the H2020 European Research Council (grant no. 766402) and the Agence Nationale de la Recherche (grant no. 18-CE09-0013).TS29

**Review statement.** This paper was edited by Geoffrey Bodenhausen and reviewed by Meghan Halse, Bernhard Bluemich, and two anonymous referees.

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

## Remarks from the language copy-editor

## Remarks from the typesetter

**TS27**    You have referred to data sets in your text. Please provide a statement on how these underlying research data can be accessed. If the data are not publicly accessible, a detailed explanation of why this is the case is required. The best way to provide access to data is by depositing them (as well as related metadata) in reliable public data repositories, assigning digital object identifiers (DOIs), and properly citing data sets as individual contributions. Please indicate if different data sets are deposited in different repositories or if data from a third party were used. Additionally, please provide a reference list entry including creators, title, and date of last access. If no DOI is available, assets can be linked through persistent URLs to the data set itself (not to the repositories' home page). This is not seen as best practice and the persistence of the URL must be secured.