# Peer review of "Simulation of NMR spectra at zero- and ultra-low field from A to Z – a tribute to Prof. Konstantin L'vovich Ivanov"

_Magnetic Resonance, 2022_

## Author Response (AR1)

Comments of the reviewers are shown in blue, our answers are in black, and citations from the revised manuscript are in green.

**Referee 1, comment RC1**

The discussion by Stern and Sheberstov introduces to readers some basic principles by which to simulate NMR spectra of simplistic molecules during free evolution of their nuclear magnetization in a zero or ultralow magnetic field (ZULF). Specifically, the compounds must contain two sets of magnetically equivalent spins-1/2, each with a different gyromagnetic ratio, for example a carbon-13 and hydrogen-1.

Technically speaking, the work is correct, but readers should note that the content is not so original, and in my opinion a narrow viewpoint on the topic of near-zero-field NMR. Butler et al. wrote a key paper back in 2013 describing the theory of zero-field NMR in not only AXn systems as presented in this work, but more complex spin systems AmXn and AXmBn as well. Many of the results were derived analytically without simulations, for example, using perturbation theory.

Simulations of zero-and-ultralow-field NMR spectra in AXn, AmXn and AXmBn are also presented in detail in several PhD theses from the early 2010s: see for example Dr Thomas Theis (2012, UC Berkeley, https://escholarship.org/uc/item/01d528kh ), Dr John Blanchard (2014, UC Berkeley, https://escholarship.org/uc/item/2mp738zn ) and Dr Tobias Sjolander (2017, UC Berkeley, https://escholarship.org/uc/item/2kj4v04n ). Readers are encouraged to consult these original sources.

We thank the Referee for suggesting these useful references. We have added this sentence to the manuscript.

"We note that several PhD theses from the Pines' group at the MIT present simulation of NMR spectra at ZULF (Theis, 2012; Blanchard, 2014; Sjolander, 2017). These theses contain code examples and are a useful resource for the beginner."

I encourage the authors to revise the paper by including more original content. For example, simulating the zero-field NMR spectra of compound that has not yet been studied experimentally, or magnetic fields bordering the zero-field condition where the perturbation theory starts to break down.

We understand that the Referee would see the paper as more relevant if new material was covered. Yet, we choose to stick to a purely educational aim. This is within the recently updated scope of *MR* (see https://doi.org/10.5194/mr-2022-18-CC5)

Alternatively, by going beyond summarizing the main results of Butler-2013 and reviewing other works where zero-field NMR spectra are calculated. I believe this would be very useful to readers interested in simulation, not only to do justice to the works listed above. Perhaps a quick way is to use a table: compound or spin system, simulation approach (e.g. exact, perturbation theory), software, literature reference.

Although we agree with the Referee that such a table would be interesting, our paper is not a review and does not claim to cover all methods for the simulation of spectra at ZULF. We only aim at showing the basics for simple examples. The use of perturbation theory is mentioned in the conclusion.

Additional comments:

- Authors and Readers should be aware of review article on zero-and-ultralow-field NMR, plus applications, published by Jiang M, Peng X et al. in 2021. This deserves to be mentioned: https://doi.org/10.1016/j.fmre.2020.12.007

**We thank the Referee for pointing out this missing reference. We have added it to the introduction.**

- Some of the equations can provide valuable physical insight into how a ZULF NMR experiment works, but it is not explained. Let us take Equation 57 as an example. The total magnetization operator Mz = gI Iz + gS Sz is applied to an eigenstate of the zero field, that is |F, mF> in the case of AXn. The result is immediately quoted in terms of Clebsch-Gordan coefficients. There is a slightly different way to write the result where the operator is given as Mz = gI (Iz + Sz) + (gS - gI) Sz. Here the first term on the right-hand side of the (=) sign leads to the eigenvalue mF gI, while the second term leads to a sum over other operators, and overall proportional to (gS - gI) times the C-G coefficient. The authors may want to mention that this second term leads to ZULF signals where the amplitude of peaks scale with (gS - gI).

**We thank the Referee for this nice suggestion, we have rewritten the paragraph around this question. The text now reads:**

"It is important to notice that, in case where  $\gamma_I = \gamma_S$ , each element of this sum becomes zero. This is shown in the Wolfram Mathematica code for all observable transitions in XA, XA2, and XA3 systems and can be rationalized in general case by the following way (see Supplement C1). The operator  $\gamma_I \hat{F}_{A,z} + \gamma_S \hat{S}_z$  (which is proportional to the initial state  $\hat{\rho}_0$ ) can be rewritten as  $\gamma_I (\hat{F}_{A,z} + \hat{S}_z) +$  $(\gamma_S - \gamma_I)\hat{S}_z$ . The first term in this expression commutes with the  $\hat{H}_{AX}$  Hamiltonian (see Eq. 48) and therefore it does not produce any observable coherences, whereas the second term does not commute with the  $\hat{H}_{AX}$  and leads to ZULF signals. "

**Referee 1, comment RC2**

In response to the Author and Community comments so far:

I think that this paper could eventually end up being a good source of reference for ZULF NMR as the authors intend, but for now it lacks in both clarity and detail.

I still find the objective unclear. The abstract promises "detail, including the tricks that are usually omitted from research papers and assuming as little prior knowledge from the reader as possible". However, the paper is not written clearly, several details are omitted, neither appear any tricks, and there are several inaccuracies (see the end) that can confuse readers regardless of their level of familiarity with theory of NMR.

Here is what I would expect the paper to contain for those entering the ZULF NMR field, or those with little experience of simulations, in this order:

 An overview of common situations where ZULF NMR simulations may be useful, maybe because analytical expressions are not available in: (a) complex spin systems, (b) free evolution near the edge of the ZULF regime, (c) evolution under pulses, e.g. dc pulses, composite pulses, (d) evolution during field sweeps, level crossings, such as those used in ZULF parahydrogen-induced polarization protocols, (e) evolution during dynamical decoupling sequences or the ZULF-TOCSY type of spin locking experiments done by the Ivanov group, (f) relaxation, (g) errors in pulse sequences or spin Hamiltonian parameters, (h) simulation of geometric effects such as sample shape, detector position, or inhomogeneous fields. The authors do not have to cover all of these in detail, but it would help to specify what they are, and give literature examples for each.

**We added the following text to the conclusions:**

"We have chosen to describe the simplest cases, i.e., experiments with thermal prepolarization with AXn systems. Using this methodology, the reader can proceed with simulating more advanced cases, where analytical solutions do not exist. This includes calculation of ZULF spectra of molecules with multiple spins (Wilzewski et al., 2017) and molecules containing three and more types of nuclei e.g. 1H, 13C, 15N, 2D (Alcicek et al., 2021); the evolution during dynamical decoupling sequences (Bodenstedt et al., 2022a), or the ZULF-TOCSY type of spin-locking experiments (Kiryutin et al., 2021), spin evolution at intermediate fields, where perturbation approaches are not valid (Bodenstedt et al., 2021). Complicated spin dynamics may occur under action of composite pulses allowing (Jiang et al., 2018; Bodenstedt et al., 2022b). The formalism we presented here is a good starting point for the description and understanding of hyperpolarized ZULF experiments e.g. those involving transfer of spin order in parahydrogen experiments at low fields. Simulations are also useful to study different kinds of imperfections such as field inhomogeneities, timing errors, etc."

• An overview of the simulation "choices" that can be made by beginners: (a) eigenvector/eigenvalue analysis in either Hilbert of Liouville space, so that for time intervals where the Hamiltonian is constant you simply take an exponential of the eigenfrequency, rather than the whole matrix, (b) use of other symmetry-adapted bases, e.g. Zeeman vs. total angular momentum, and treat smaller subspaces one by one, (c) brute-force propagation of the Liouville von-Neumann or Schrodinger equation, say in the Zeeman basis. In the manuscript, it appears that a combination of (a) and (c) has been used, but it is not completely clear when and how.

**We added the following text:**

"The brute force calculation of the exponential operator in an arbitrary basis is computationally challenging as it requires calculating the Taylor expansion of the  $\hat{H}$  operator. To avoid this, the calculation of the propagator (Eq. 27) is usually performed by diagonalizing the Hamiltonian and then taking the complex exponent for each of its eigenvalues,  $\exp(-i\omega_k t)$ , where  $\omega_k$  denotes the  $k^{\text{th}}$  eigenvalue. Therefore, the transformation to the eigenbasis of the Hamiltonian implicitly happens during most spin dynamics simulations, meaning that, even if it was not set by the user, this is likely done by the linear algebra packages of the software. One may note that the basis does not affect the result of the calculation but the choice of a more appropriate one may help rationalize the problem. In many cases, the initial choice is the Zeeman basis, in which spin operators are readily introduced based on Kronecker products of the Pauli matrices. Depending on the symmetry of the problem it might be more convenient to change the basis to another one. As we will see in section 4.1, a choice of coupled basis is preferable for understanding zero-field *J*-spectroscopy of coupled spins."

A few illustrative examples from the list (a)-(g) above. Novelty is not required if these examples reproduce the results of previous work with appropriate citation. Such as "In work (author et al., 20xx) this was shown. We will now show the reader how this can be simulated by ..." and then proceed to do so, step by step, without leaving any gaps or diverting the reader to sources elsewhere. I will concede that the examples given by the authors are acceptable for a tutorial paper, but they do lack the complete start-to-finish detail that would allow a graduate student

**with a basic knowledge of quantum mechanics to sit down at a computer and work through them without getting stuck.**

We thank the Referee for this criticism but we disagree. Indeed, the paper considers only basic examples but it provides a lot of details as well as link to step-by-step simulation code. We think that having these hints reader will be able to implement acquired skills to perform more advanced simulations.

In my opinion, the authors are assuming a lot more of the reader than they realize, so their work can only be followed by someone who already has an expert level of experience in spin dynamics simulations for NMR, and who is probably capable of looking up the PhD theses already mentioned in the previous review comment.

Below are some specific or technical comments that I recommend the authors address:

• Line 21: "Use frequency dispersion" instead of "chemical shift dispersion"

**We have modified the text according to the Referee's suggestion.**

• Line 24: The reference "(Thayer and Pines, 1987)" does not seem an appropriate citation with the 1.2 GHz NMR magnet?

**We have fixed the position of the references.**

• Line 29: "Mumetal", rather than "μ-metal"

**We have modified the text according to the Referee's suggestion.**

• Line 31: "does no longer influence the outcome of the experiment" is highly ambiguous. Consider replacing with "where the Larmor period is much longer than the coherence time, so that the field can be completely neglected".

**We have modified the text to:**

"In this paper, 'Zero-field' (ZF) designates the regime where heteronuclear spin-spin interactions dominate over spin-field interactions (Zeeman interactions) and the residual spin-field interactions are small enough for the Larmor period to be much longer than the coherence time (Blanchard and Budker, 2016). When this condition is met, decreasing the residual field to even lower values leaves the NMR spectrum unchanged."

Line 47: "at ZULF, there is no such intuition as the vector model". Is that really true? The AX spin system can often partitioned into an isolated 2LS where cyclically commuting operators can be represented as a vector model picture, see for example https://doi.org/10.1063/PT.3.1948 or https://doi.org/10.1021/acs.jpca.6b04017 supporting information.

**Our statement was indeed inappropriate. To avoid confusion, we replaced this sentence by:**

"At ZULF, couplings between spins need to be taken into account even to describe the simplest experiment, which consists of detecting the coherence between the singlet  $S_0$  and triplet  $T_0$  states of a pair of *J*-coupled heteronuclei, e.g. 1H and 13C."

• Line 56 and 77: "using a theoretical framework coming from atomic physics". I don't think this is ok. You apply the rules of addition of angular momentum. This is not specific to atomic physics – it is just as widely used in molecular (e.g. rotational) spectroscopy and NMR.

**We agree and changed the sentence to be:**

"The formalism of addition of angular momenta (widely used in atomic physics, and rotational spectroscopy, but less frequently in liquid-state NMR) can therefore be used to describe ZULF experiments."

• Line 69: "we assume the reader is familiar with general concepts of NMR but not necessarily with simulation". This is not a clear sentence.

**The sentence was changed to:**

"We assume that the reader is familiar with general concepts of NMR but that they are not necessarily used to perform spin dynamics simulation."

 Line 82: Spinach already has a large set of examples for zero-field NMR, e.g., http://spindynamics.org/wiki/index.php?title=Zerofield.m and http://spindynamics.org/wiki/index.php?title=Zulf\_abrupt.m (sudden field drop), and probably SpinDynamica does too. These could be mentioned more directly, even though the above links are not permanent ones.

**We have added a direct reference to the link proposed by the Referee.**

"The people who have programmed these have already gone through the hurdles of making them efficient and versatile for us and even provide code examples for the simulation of NMR spectra at ZULF.1"

**1 See for example http://spindynamics.org/wiki/index.php?title=Zerofield.m**

• Line 269: "is sometimes referred to as the sandwich formula" needs a reference. The only time I have read about "sandwich formulae" in the context of NMR is the rotation sandwich formulae in Levitt's book "Spin Dynamics: basics of nuclear magnetic resonance". These are specific to cyclically commuting triads of operators, i.e. fictitious spin-half representation of a two-level subspace

**We thank the Referee for pointing out this improper terminology. We have removed any reference to the sandwich formula.**

• Line 272: the "propagation operator" is a propagator

**We have removed the expression "propagation operator"**

• Starting line 341: For the sake of clarity, let us not call signal or time a vector. Perhaps "list" or "array" of time points or time-amplitude pairs

We have modified the text according to the Referee's suggestion but indicating that this type of array is called a vector in MATLAB's programming environment. The next now reads:

"Let us call t and S the arrays of numbers containing the time and corresponding time domain signal values, respectively, which resulted from the previous steps (note that, in MATLAB's programming environment, such arrays are usually called vectors)."

• Line 505- 507: The phenomenon described is not nutation. Better to refer to the excitation curves as "Rabi oscillation curves"

We have modified the text according to the Referee's suggestion. See section 3.2.

• Style comment: please cite equations as "..., see Equation (x)" rather than "see (x)".

We adapted the formatting according to the Referee's suggestion.

**Referee 2 – Bernhard Blümich, comment RC3/CC3**

**General comment:**

I really enjoyed reading this paper. It reminded me of my time as a PhD student, when I tried to understand the density matrix formalism and program the transverse magnetization response to some odd excitation using assembler code. In those days long gone I found the then recent paper by P.D. Buckley, K.W. Jolley, D.N. Pinder "Application of density matrix theory to NMR line-shape calculations", PNMRS 10 (1975) 1-26 most helpful as it gave hands-on examples which I could adapt to my own case of interest. Compared to that old paper, the manuscript at stake is written even more in a tutorial style, working out the details of the A3X system as an example. In my view this is a valuable guide to beginning PhD students interested in ZULF NMR. Clearly this manuscript is not a review, nor does it cover all common cases encountered with ZULF NMR, but I find it to be quite useful as a starting point for one's own simulations. I recommend it to be published following revision. In particular, the corrections and comments voiced in the discussion so far should be considered and implemented so long as the length of the manuscript can largely be maintained.

We thank the Referee for this very positive feedback on our manuscript. We would also like to thank him for the detailed comments, which will indeed improve the readability for the beginner.

**Detailed additional comments:**

Line 20 ff: "Increasing magnetic field strength boosts the sensitivity thanks to higher Boltzmann nuclear polarization and higher Larmor frequency". Although stated many times in the literature, this only applies to high-resolution spectroscopy at constant linewidth in frequency units. It is the peak amplitude and not the peak integral in the spectrum that defines the sensitivity. (This raises the question of the "homogeneity" of the zero-field and its impact on spectral resolution and sensitivity at a given polarization.) Perhaps one can write "increasing the field strength while maintaining the linewidth ...".

We thank the reviewer for pointing out this imprecision. We have modified the text as follows:

"NMR spectroscopists know well the advantages of performing experiments at the highest possible magnetic field. Increasing magnetic field strength boosts the sensitivity thanks to higher Boltzmann nuclear polarization and higher Larmor frequency (provided the signal linewidth is maintained constant)."

**Line 133: Figure 1B**

The reference was updated

Line 170: What defines the directions of the axes at zero field?

Several conventions exist. We have added a sentence to the make it clear to the reader:

"We assume that the OPM is configured so as to be sensitive to magnetic field along the *z*-axis and the spins are initially prepolarized along the same axis. Defining this axis as *z* is a natural choice for high-field NMR spectroscopists but note that other conventions exist (see for example Ref. (Ledbetter et al., 2011))."

**Lines 224 and 226: Are "density operator" and "density matrix" used synonymously? That is confusing to the beginner.**

Indeed. We have rewritten the section on the density operator to make it clear that the density matrix is a possible representation of the density operator. The paragraph now reads

"The state of a spin system during an NMR experiment is described by a density operator. If  $|\psi\rangle$  is a ket representing the state of the system as a linear combination of basis states (like those defined in Eq. 1 and 9), the density operator is given by

 $\hat{\rho} = \overline{|\psi\rangle\langle\psi|},\tag{17}$

where the upper bar represents the ensemble average over all identical spin systems in the sample – the operation performed by the density operator. This averaging makes the density operator formalism well-suited for NMR where the experiment consists of observing a large number of identical spin systems at the same time, rather than a single spin system. The matrix representation of the density operator (and of any other spin operator) is achieved by calculating all the matrix elements  $\rho_{rs} = \langle r | \hat{\rho} | s \rangle$ , where  $| r \rangle$  and  $| s \rangle$  are basis states."

**From this point in the text, we only refer to the density matrix.**

Equation (19) and throughout the entire manuscript: The format of constants, variables, functions is inconsistent. This poses extra barriers for a student struggling to understand the math. The formatting rules apply independently to the quantity under consideration and its superscripts and subscripts. For example, the subscript "eq" in (19) is not a variable and should not be written italic.

We have gone through the manuscript and attempted to correct all such occurrences. We hope we haven't missed any!

Lines 335: What is a "Fourier transform function"? The Fourier transformation is an operation, the result of which is a transform. Both, input and output of the Fourier transformation are functions. See also line 345.

The word "function" refers to the programming object which performs the Fourier transform. To avoid confusion, we have replaced "fast Fourier transform functions" by "functions for fast Fourier transformation".

Lines 399, 416 "... multiply the frequency domain signal ...." Should probably read "... multiply the abscissa of the frequency-domain signal ...".

**We have implemented the proposed change.**

Line 435, "Hermitian": Charles Hermite was a French mathematician. Consequently, the attribute referring to his name is correctly spelled "Hermitean". Admittedly, contrary to the older literature, one often finds "Hermitian" in the modern literature. You may want to pay tribute to the correct spelling of his name in the manuscript.

**We have implemented the proposed change.**

Caption to Fig. 5 and elsewhere in the text: "with a zero-filling of 65'536 points" should be replaced by "with zero filling to 65,536 points", because you did not fill in 65,536 zeroes.

**We have implemented the proposed change.**

Overall the manuscript is well written, with just a few language issues , which I am sure, the Copernicus editors will pick up.

**Referee 3 – Meghan Halse, comment RC4**

This is a well written and clearly presented tutorial paper. I think that a pedagogical paper supported by a well-documented and accessible simulation code to bridge the gap between a basic NMR understanding using the vector picture and advanced product operator simulation packages like SPINACH and Spin Dynamica will be of benefit to the NMR community. This paper goes a long way to providing this link; however, I think that focus on ZULF NMR simulations, particularly in the earlier sections of the paper, may be confusing for many non-experts and may therefore limit the audience for this very nice contribution. The final section of the paper is quite theoretically challenging for the non-expert and will likely only be of interest to those in the ZULF community. Overall I think that this paper is publishable with minor revisions; however, the addition of some clearer links to high-field NMR simulations (see specific suggestions below) along with a slightly modified version of the MATLAB code to allow the user to perform a more familiar HF NMR simulation to compare with the ZULF simulations would make this a much more broadly useful contribution.

We would like to thank the Referee for this careful read through of our manuscript. There were indeed missing definitions and the Referee's suggestion on where to add these definitions were very welcome.

We modified the manuscript to make a clearer comparison with high-field NMR. In Sections 2.1 to 2.7, where we present the theory of the numerical spectrum simulation at ZULF, we made it clear what differs in simulating a spectrum at high field. We have also added further details to Section 2.8, which compares high field and ZULF NMR. Finally, we have added a MATLAB script and the associated PDF (Supplement B2) to present in great detail a spectrum simulation at zero field, ultra low-field, and high field (both for 1H and 13C spins). We hope that the comparison between high field and ZULF will appear satisfactory to the Referee.

**Specific Comments:**

There are a few concepts that are mentioned by not defined clearly and so may confuse a less knowledgeable reader

**Hilbert space (p6 line 154) is mentioned but not explicitly defined**

We have rewritten the paragraph following Eq. 1 and added the sentence below. We chose not to give a formal definition of Hilbert spaces to avoid confusing the reader with details that are not necessary for to understand the process of the simulation.

"The space spanned by these two vectors is called a "Hilbert space" and has dimension 2, as indicated by the superscript in B\_Z^2."

Bra-ket notion is used in eq. 17 and 18 without being explained/introduced.

On a related note it would be helpful to define the alpha and beta kets with the matrix notation when they are first introduced. I don't think it is made quite clear the relationship between these states and the columns/rows of the matrices.

We have rewritten the text around Eq. 1 and included the definition of the ket and the matrix representation of the ket.

"The state of any spin system can be represented as a linear combination of basis vectors, which are called "kets" in Dirac's notation and are represented by the symbol  $|\rangle$ . For a single spin-1/2, two basis kets are necessary to represent the state of the system. We chose to represent the spin system in the Zeeman basis

$$\mathcal{B}_{Z}^{2} = \{ |\alpha\rangle, |\beta\rangle \} = \left\{ \begin{pmatrix} 1\\ 0 \end{pmatrix}, \begin{pmatrix} 0\\ 1 \end{pmatrix} \right\}.$$
(1)

The  $|\alpha\rangle$  and  $|\beta\rangle$  states correspond to the spin being parallel and antiparallel with the magnetic field, respectively. The general state in which the spin may be found is a linear combination of these two basis states. Because these states and their associated kets form a basis, their vector representation have the canonical form with only 0 and 1 coefficients."

P8 line 201 is the first explicit mention of eigenstates. I think it would be useful when introducing the Zeeman Hamiltonian (eq 5) to explicitly define the alpha and beta kets as the eigenstates and to show the relationship between these states and the matrix representation. This would also be an opportunity to introduced the time-independent Schrodinger equation for 1 spin before the introduction of the Liouville von-Neuman equation for the density matrix (see point 3 below).

We have added the sentence below in the paragraph following Eq. 5. However, we prefer not to define introduce the Liouville-Von Neumann equation before we introduce the density matrix.

"The Zeeman states,  $|\alpha\rangle$  and  $|\beta\rangle$ , which correspond the spin being parallel and antiparallel with the magnetic field, respectively, are eigenstates of the Zeeman Hamiltonian, i.e., they satisfy the relations  $\hat{H}_{\rm Z}|\alpha\rangle = +1/2|\alpha\rangle$  and  $\hat{H}_{\rm Z}|\beta\rangle = -1/2|\beta\rangle$ . Eigenstates of a Hamiltonian are of particular importance; they are states which do not evolve under the effect of that Hamiltonian (ignoring the accumulation of the phase factor, which turns out to be irrelevant in most of the experiments), i.e., stationary states."

Eq. 11 – the use of Ix, Iy and Iz to denote the sum of the anguluar momentum operators over a number of spins is confusing as this same notation is used in eqs. 3 and 13 to denote just the 2x2 angular momentum matrices. I suggest using a different letter such as L to denote a sum over multiple spins. I had a similar issue with the definitions in eqs 48-51, where I was unsure of the definitions of various terms and found the use of bold and italics unclear/inconsistent.

Eq. 11 comes from M. Levitt's Spin Dynamics (Eq. 14.13). We have added a sentence to point out the possible confusion to the reader:

"Note that these operators are represented by the same symbol as their equivalent in the single-spin Hilbert space (see Eq. 3). It should be clear from the context whether the operator corresponds to a single-spin or multiple-spin Hilbert space."

However, we have changed some of the variable names in Section 4 to avoid confusion.

As for the bold and italics in Eqs. 48-51, thank you for noticing it, as there were problems with formatting. Now, italics is used everywhere and bold to denote vector operators. We also add a reminder that

"*m**I* and *m*S are the z-projection of total spins *I* (for n protons  $I \in \{-\frac{n}{2}, -\frac{n}{2}+1, ..., \frac{n}{2}-1, \frac{n}{2}\}$ ) and z-projection of S (in case if S is carbon-13,  $S \in \{-\frac{1}{2}, \frac{1}{2}\}$ ) respectively."

Eq. 23 for the equilibrium matrix includes implicitly within it a two-spin-order term that emerges from the product. It is common in NMR textbooks, when analysing pulse sequences using product operators to express the equilibrium starting state as just the sum of the Iz operators for the various spins of interest and to omit any higher spin order terms. I think it would be useful to explicitly show the expansion and explain why, and under what circumstances, the single spin operators are a reasonable approximation of the equilibrium density matrix at thermal equilibrium.

**We have shown the expansion in Eq. 23 stating the limit in which the approximation is valid. The text now reads**

"For a *n*-spin system, we take the Kronecker product of density matrices of individual spins  $\hat{
ho}_{{
m ea}.l}^{2 imes2}$

$$\hat{\rho}_{eq} \approx \bigotimes_{l=1}^{n} \hat{\rho}_{eq,l}^{2\times2} = \bigotimes_{l=1}^{n} \left( \frac{\hat{1}^{2\times2}}{2} + P_l \hat{I}_z^{2\times2} \right) = \frac{\hat{1}}{2^n} + \frac{1}{2^{n-1}} \sum_{l=1}^{n} P_l \hat{I}_{lz},$$
(23)

The expression is approximate in the sense that it neglects all spin-spin interactions. This approximation is valid unless the system is highly polarized, which is the case even at very high field (without hyperpolarization). To avoid confusion, we specified that the operators  $\hat{\rho}_{eq,l}^{2\times2}$ ,  $\hat{1}^{2\times2}$ , and  $\hat{I}_{z}^{2\times2}$  act on a single-spin Hilbert space (2 × 2 matrix). On the contrary, the operators  $\hat{1}$  and  $\hat{I}_{lz}$  act on spin states of *n*-spins, and accordingly their matrix representations have dimensionality of  $2^n \times 2^n$  (for spins-1/2). As shown by Eq. 23, one may compute the density matrix either using the Kronecker product of operators in a single-spin Hilbert space or by summing the operators in a Hilbert space of *n*-spins."

I think it would be useful prior to Eq. 26 to explicitly give the Liouville-von Newumann equation to which it is a solution.

**We have implemented the proposed change.**

On p14 line 350 T2 is defined as the "coherence time constant", which is a true definition but will be unfamiliar to most readers who will better know this as the spin-spin relaxation or transverse relaxation time constant.

We have added the sentence below after introducing the coherence time constant.

**"Note that the coherence time constant is often referred to as the spin-spin relaxation constants or transverse relaxation time constant."**

In section 2.7, the focus is put on the acquisition parameters for ZULF spectra without any discussion of how these relate to the acquisition parameters for standard HF spectra, where the spectrum is acquired in the rotating frame and Larmor frequencies are defined as chemical shift offsets relative to a reference frequency. As the ppm scale and HF NMR is the most natural reference point for most readers, I think it is important to describe the two regimes and the relationship between them. Indeed, that is what I was expecting in the "comparison with high-field NMR" section.

We have added details along Section 2 to describe how the process is different at high field and at ZULF. We have also added Supplement B2 to show a side-by-side ZULF and high-field simulation. The high-field simulation is done in the lab frame and the FID is "demodulated" after the propagation. The process is explained in Supplement B2.

Due to the focus on ZULF NMR the non-standard case of static field pulses is introduced but the standard representation of ideal RF pulses in the high field regime using rotation operators is not described. The first mention of a rotation operator is on line 599 (p26). This is potentially confusing as the role of rotation operators in NMR simulations has not been described previously. I think it would be helpful in the pedagogical spirit of this paper to include a brief description of this in the theory section.

We have added the following paragraph to introduce the rotation operators after introducing the solution to the Liouville-Von Neumann equation.

"An important case of propagator is the rotation operator. For an angular momentum operator  $\hat{I}_{\mu}$ , with  $\mu \in \{x, y, z\}$ , the propagator  $\exp(-i\hat{I}_{\mu}\theta)$  is called a rotation operator; it represents a rotation of the spins of angle  $\vartheta$  around axis  $\mu$ , when applied to the density matrix using Eq. 26. For a single spin subject to a static magnetic field along the *z*-axis, the total Hamiltonian is the Zeeman Hamiltonian (see Eq. 5) which causes the spin to rotate around *z*-axis; this rotation can be expressed using the rotation operator  $\exp(-i\hat{H}t) = \exp(-i\omega_0\hat{I}_z t)$  with angle  $\omega_0 t$ ."

In the introduction to ZULF NMR the authors choose to define ULF NMR as where there is a Zeeman contribution but this is not dominant. This definition excludes NMR in the tens of uT regime, notably Earth's field NMR. By excluding EFNMR a key step in the development of ZULF NMR is omitted, from the initial EFNMR experiments by Packard and Varian in 1954 (Phys Rev), where pre-polarisation and non-adiabatic field switching was first used to the first pulsed EFNMR work of Callaghan and LeGros (Americal Journal of Physics, 1982) and up to the work by Appelt et al (Nature Physics 2006), who was the first to introduce the idea of using a Halbach for prepolarisation at a few Tesla before detection in uT fields.

We have added the definition of EF-NMR below to the introduction for completeness and clarity. However, we prefer not to enter the historic details (although interesting) to avoid giving the reader too much information that is not directly necessary to the understanding of the case presented in the paper.

"The regime where the intensity of heteronuclear spin-spin interactions is on the order of that of the spin-field interactions occurs typically in the range of  $\mu$ T to tens of  $\mu$ T and is referred to as Earth-field NMR (EF-NMR) (Callaghan and Le Gros, 1982; Appelt et al., 2006)."

**Technical corrections**

I noted a couple equation references that appear to be incorrect:

I think on p17 line 411 it should be eq 36 while line 412 should be eq 37.

I think on p18 line 456 it should be eq 35.

We thank the Referee for pointing out these mistakes. The equation numbers have been updated.

**Referee 4, comment RC5**

The work from Q. Stern and K. Sheberstov is a tutorial paper addressed to PhD students approaching NMR experiments simulations at ultra-low or zero field.

Despite the manuscript is well written and over all correct, I struggle to see a real pedagogical aim in this work. In my honest opinion, I doubt that first year PhD students will be able to open their MATLAB and start to run NMR simulation thanks to this manuscript. I must agree with Reviewer 1 about the fact that the authors assume a lot more of the readers' pre-knowledge than what they think. Through the text, apart some NMR theory that can be found in any topic specific textbook (e.g. Spin Dynamics from Prof Levitt), I could not see any real trick that would "carry by hand" the newbie from analytical formulas to computation. Moreover, especially at the beginning, jumping straight to ZULF simulation might be confusing for the non-experts. The paper could be useful for a broader audience if a "standard/high-field" NMR section was added.

The following is just a suggestion, but, why not starting with a very simple system (non-interacting spins ½ at thermal equilibrium with Zeeman interaction only, no propagation of the density matrix yet just CW style) and show step-by-step how to go from the Hamiltonian to the spectrum in MATLAB, reporting even chunks of code into the main text. Then, we can add the J-coupling and see how the spectrum changes, propagate the density matrix etc.

We modified the manuscript to make a clearer comparison with high field NMR. In Sections 2.1 to 2.7, where we present the theory of the numerical spectrum simulation at ZULF, we made it clear what differs in simulating at high field. We have also added further details to Section 2.8, which compares high field and ZULF NMR. Finally, we have added one script and the associated PDF (see Supplement B2) to present in great detail a spectrum simulation at zero field, ultra low field, and high field (both for 1H and 13C spins). We hope that the comparison between high field and ZULF will appear satisfactory to the Referee.

As a first year PhD student I would love to find in the literature something like this!

Below some minor details:

Line 27: the second "interaction" is redundant

This sentence was rewritten. It now reads

"In this paper, 'Zero-field' (ZF) designates the regime where heteronuclear spin-spin interactions dominate over spin-field interactions (Zeeman interactions) and the residual spin-field interactions are small enough for the Larmor period to be much longer than the coherence time"

Line 41: add a reference after "coils" about OPM detection for ZULF

We have added a reference to Ledbetter et al. 2009.

Line 42: I would remove "This simple idealization of"

We have replaced "This simple idealization" by "The representation". The sentence now reads

"The representation of a single spin system as a vector in 3D-space is a powerful tool [...]"

Line 427: "positioned" instead of "position"

We have implemented the suggested change.